# Trace Element Geochemistry of Alluvial TiO₂ Polymorphs as a Proxy for Sn and W Deposits

**Miguel Gaspar** [1,2,*] , **Nuno Grácio** [2,3] , **Rute Salgueiro** [3] **and Mafalda Costa** [4]

1   Departamento de Geologia, Faculdade de Ciências da Universidade de Lisboa, Ed. C6, Piso 4, Campo Grande, 1749-016 Lisboa, Portugal
2   Instituto Dom Luiz (IDL), Faculdade de Ciências da Universidade de Lisboa, Ed. C1, Piso 1, Campo Grande, 1749-016 Lisboa, Portugal
3   Laboratório Nacional de Energia e Geologia (LNEG), Mineral Resources and Geophysics Research Unit, Ap. 7586, 2611-901 Amadora, Portugal
4   HERCULES Laboratory, University of Évora, Palácio do Vimioso, Largo Marquês de Marialva 8, 7000-809 Évora, Portugal
*   Correspondence: mgaspar@fc.ul.pt

**Abstract:** The Segura mining field, the easternmost segment of the Góis–Panasqueira–Segura tin–tungsten metallogenic belt (north–central Portugal), includes Sn-W quartz veins and Li-Sn aplite-pegmatites, which are believed to be genetically related to Variscan Granites. Sediment geochemistry indicates granite-related Ti-enrichments, locally disturbed by mineralization, suggesting magmatic and metamorphic/metasomatic titaniferous phases. Therefore, Segura alluvial samples and the geochemistry of their TiO₂ polymorphs (rutile, anatase, and brookite) were investigated, and their potential as exploration tools for Sn and W deposits was evaluated. The heavy-mineral assemblages proved to be good proxies for bedrock geology, and TiO₂ polymorph abundances were found to be suitable indicators of magmatic and/or metasomatic hydrothermal processes. The trace element geochemistry of Segura's alluvial rutile, anatase, and brookite is highly variable, implying multiple sources and a diversity of mineral-forming processes. The main compositional differences between TiO₂ polymorphs are related to intrinsic (structural) factors, and to the P-T-X extrinsic parameters of their forming environments. Anomalous enrichments, up to 9% Nb, 6% Sn and W, 3% Fe, 2% Ta, and 1% V in rutile, and up to 1.8% Fe, 1.7% Ta, 1.2% Nb, 1.1% W 0.5% Sn and V in anatase, were registered. Brookite usually has low trace element content (<0.5%), except for Fe (~1%). HFSE-rich and granitophile-rich rutile is most likely magmatic, forming in extremely differentiated melts, with Sn and W contents enabling the discrimination between Sn-dominant and W-dominant systems. Trace element geochemical distribution maps show pronounced negative Sn (rutile+anatase) and W (rutile) anomalies linked to hydrothermal cassiterite precipitation, as opposed to their hydrothermal alteration halos and to W-dominant cassiterite-free mineralized areas, where primary hydrothermal rutile shows enrichments similar to magmatic rutile. This contribution recognizes that trace element geochemistry of alluvial TiO₂ polymorphs can be a robust, cost- and time-effective, exploration tool for Sn(W) and W(Sn) ore deposit systems.

**Keywords:** TiO₂ polymorphs; alluvial; heavy minerals; Sn-W deposits; trace elements; geochemical footprints; geochemical fingerprints; geochemical exploration

## 1. Introduction

Trace-element geochemistry is one of the most widely used geochemical tools for tracing geological processes and obtaining information about the origin and genesis of various types of rocks and minerals [1]. Some of these processes result in economic concentrations of metals, leading to mineral deposits. Trace element distribution in mineralized rocks and corresponding geochemical alteration halos, is nowadays critical for understanding the

mineralizing processes and for evaluating the economic potential of a particular occurrence or region.

The study of heavy minerals (d > 2.8–2.9 g/cm$^3$) [2] has been successfully applied to mineral exploration for centuries, dating back to 300 BC [3]. The first studies used minerals' physical properties to identify specific pathfinder minerals and to characterize the mineral assemblages of heavy mineral (HM) concentrates, e.g., [4–6]; this became a systematic exploration tool since the 20th century [7,8]. New analytical techniques have also improved the ability to use HM chemical and isotopic compositions in geochemical exploration [8–48].

Titanium dioxide (TiO$_2$) polymorphs (rutile, anatase, and brookite) are common accessory minerals that fall into the heavy mineral group. The geochemical affinity of Ti with HFSE (Nb, Ta, Zr, Hf) and other transition and base metals (Fe, Cr, V, Sn, Sb, Mo, W) makes TiO$_2$ polymorphs potential minerals to exhibit anomalous metal enrichments related to ore-forming systems. In the last two decades, this potential has been incrementally explored, especially in rutile, the most stable TiO$_2$ mineral phase in most geological environments, enabling the characterization and discrimination of several mineralizing systems [49–73]. Rutile from various ore deposit types has been reported to exhibit significant enrichments in Sn, W, Nb, Ta, Sb, Cr, and V, while anatase and brookite commonly display low trace element contents [68].

Clark and William-Jones, in their comprehensive study on rutile geochemistry in 26 mineral deposits of various natures [57], reported high concentrations of W, Sn, Nb, and Ta in rutile from hydrothermal Sn-W veins and rare-element pegmatites associated with granites. Carocci et al. [70], analyzed rutile from the mineralized veins and host rocks of the Panasqueira Sn-W-Cu deposit, and reported enrichments in W and Sn (Nb, Ta), considering them good pathfinders for W deposition. The abundance of W-enriched rutile in the tourmalinized host rocks of the Panasqueira deposit seems to be a marker of the onset of the main wolframite depositional event [70].

The currently inactive Segura mining field, corresponding to the easternmost segment of the Góis–Panasqueira–Segura Sn–W metallogenic belt (north-central Portugal), contains several cassiterite and wolframite quartz veins, as well as Li-Sn aplite-pegmatites, which are believed to be genetically related to the Variscan Segura granitic massif. Sediment geochemistry surveys carried out by LNEG for the region [74], suggest zoning and Ti enrichment related to granite emplacement, locally disturbed by mineralized bodies, pointing towards the development of metamorphic/metasomatic titaniferous phases.

This contribution reports a HM study on alluvial samples from this region, and a geochemical characterization of their TiO$_2$ polymorphs, to investigate potential exploration tools for Sn and W deposits. Heavy minerals are highly resistant to weathering and can physically separate from rocks and accumulate in alluvial deposits, forming detrital anomalies and natural HM concentrates. Considering that TiO$_2$ polymorphs are accessory minerals, the use of alluvial samples has the advantage of reducing drastically the amount of rock that needs processing and, consequently, the time and budget spent in HM separation. Moreover, this study shows that alluvial HM assemblages echo the bedrock geology of the catchment area, including ore bodies, and that abundance distribution maps of alluvial TiO$_2$ polymorphs, and other pathfinder minerals, can pinpoint a diversity of Sn and W mineralization styles. In this study, we applied electron-probe micro analysis (EPMA) to characterize the chemical compositions of alluvial TiO$_2$ polymorphs. Rutile and, to some extent, anatase show variable but significant enrichments in Sn, W, Nb, Ta, V, and Fe in samples typically derived from the mineralized areas. We learned that if proper attention is paid to the intrinsic and extrinsic factors that control trace element substitution for Ti, to a correct mineral identification, and to the possible multi-source populations that characterize alluvial samples, the trace element geochemistry of alluvial TiO$_2$ polymorphs can be a robust exploration tool for Sn-W deposits.

## 2. Geological Setting

This study was performed in alluvial samples collected in the Segura region, sited in Castelo Branco district (central Portugal), near the Spanish border (Figure 1). To support the HM analyses and geochemistry discussed in this paper, a brief geological framework is outlined below.

The Segura region is located within the Central Iberian Zone (CIZ), a geotectonic unit that forms the central part of the Iberian Variscan orogenic belt (Figure 1a). The main outcropping lithologies include: metasedimentary rocks of the Beiras Group belonging to the Dúrico-Beirão Supergroup [75], or to the ante-Ordovician Slate-Greywacke Complex (SGC), as it is traditionally known [76]; granitic rocks belonging to the Segura Massif (SM); and several dykes, veins, and masses of different natures [77,78]. The metasedimentary sequence corresponds to intercalations of Neoproterozoic to Lower-Cambrian greenschist facies metagreywake and metapelite rocks, occasionally interbedded with metacoglomerate layers [77,78].

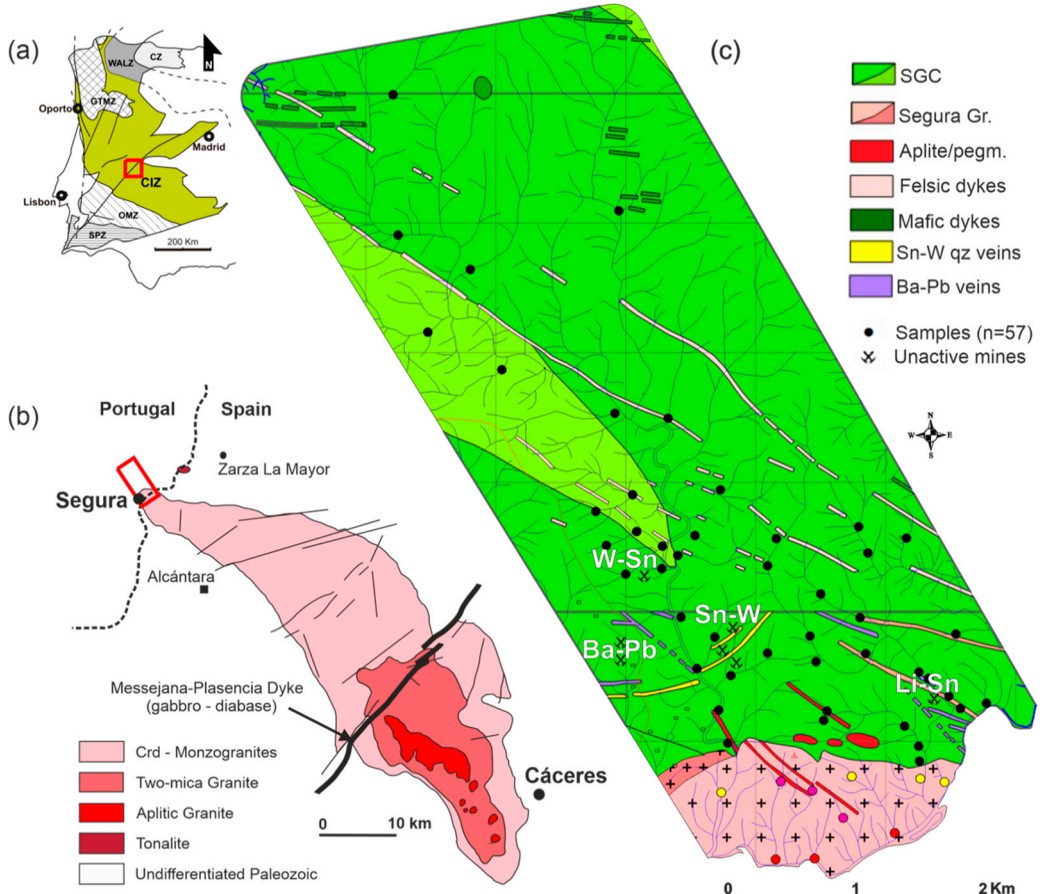

**Figure 1.** Geological setting and sample location, adapted from [37]. (**a**) Geotectonic map of the Iberian Variscan orogenic belt, adapted from [79]. CZ—Cantabrian Zone; WALZ—West Asturian-Leonese Zone; GTMZ—Galicia Trás-os-Montes Zone; CIZ—Central Iberian Zone; OMZ—Ossa Morena Zone; SPZ—South Portuguese Zone. (**b**) Cabeza de Araya Batholith, adapted from [80]. Crd—cordierite. (**c**) Geology of the studied area, adapted from [77,78]; and extracts from vectorial format Geological sheets n° 283 Salvaterra do Extremo, scale 1:25,000, 1996, and n° 295 Segura, scale 1:25,000, 2010, LNEG with alluvial sample location. SGC—Slate Greywacke Complex; SM—Segura Massif (1—two mica granite; 2—muscovite granite). The areas corresponding to the main Sn-W, W-Sn, Li-Sn, and Ba-Pb orebodies are highlighted on the map.

The general regional structure shows preferential WNW-ESE orientation and is characterized by folding of the Beiras Group metasediments, with the WNW-ESE fold axes dipping to SE and a penetrative axial plane cleavage, resulting from the first Variscan deformation phase (D1) [78]. A dense fault network, developed during the third Variscan deformation phase (D3) owing to dominant brittle behavior, is later reactivated throughout the Alpine cycle [78].

The main granitic body in the region, known as the Segura Massif (SM), corresponds to the NW tip of the Cabeza de Araya Batholith, which extends from the vicinity of Cáceres (Spain) to Segura (Figure 1b). This elongated elliptical batholith, with a NW–SE-oriented major axis, is a complex, composite, and zoned granite, with two distinct facies recognized in the studied area (Figure 1b): an internal coarse-grained two-mica granite bordered by a medium-to fine-grained muscovite facies [78,81]. The two-mica granite has a granular, locally serial hypidiomorphic texture, consisting of quartz, K-feldspar, albite, biotite, chlorite, muscovite, apatite, tourmaline, zircon, sillimanite, and rutile [78,81]. The muscovite granite presents a granular hypidiomorphic texture, dominated by tabular crystals of muscovite that occur together with quartz, K-feldspar, albite, rare biotite, chlorite, apatite, zircon, rutile, and souzalite [81]. Several aplite–pegmatite structures occur in association with this batholith (Figure 1c). SM is also crossed by the lengthy NE–SW Messejana–Plasencia Dolerite Dyke (Figure 1b) and a NW–SE garnet-cordierite granite porphyry dyke [82]. The Segura Massif falls into the syn-to-late-D3 Variscan granites yielding zircon and monazite U-Pb crystallization ages of $311.0 \pm 0.5$ Ma for the two-mica granite and $312.9 \pm 2.0$ Ma for the muscovite granite [83]. These granites developed a contact metamorphic halo more than 500 m long, divided into two zones: a 20 m proximal zone, dominated by cordierite- and sillimanite-bearing hornfels, and a more distal envelope, characterized by mica schists with cordierite porphyroblasts [81].

The mineral occurrences in this study area are lumped altogether as the Segura Mining Camp (SMC), the easternmost segment of the Góis–Segura tin-tungsten metallogenic belt [84]. This W–E belt, which is approximately 8000 km$^2$ long, is acknowledged for its W and/or Sn deposits, embracing, from west to east, the Vale Pião (W-Sn), Panasqueira (W-Sn-Cu), Argemela (Sn), and Segura (Sn-W-Ba-Pb-Zn) mines.

The SMC, or Segura mine includes Sn-W quartz veins and Li-Sn-bearing aplite-pegmatite dykes, which are believed to be genetically related to the Variscan Segura granites, and later Alpine Ba-Pb-Zn quartz breccia veins [85]. These deposits were exploited between 1942 and 1953 by the Empresa Mineira de Segura, Lda, which produced approximately 100 tons of cassiterite, 12 tons of wolframite, 525 tons of baryte, and 211 tons of galena [85].

The quartz veins, with variable thicknesses (from a few centimeters to a few meters), are emplaced and controlled by the main structural features of the region [78]. Some of the veins are subvertical, parallel to the main regional fault orientations, and may or may not be mineralized. Cassiterite and/or wolframite mineralization is mostly found in sub-horizontal vein structures, not exceeding 10 cm in average thickness, and believed to represent dilatational fissures [78]. The sub-horizontal quartz veins present a granular xenomorphic texture and are mostly composed of quartz, muscovite, cassiterite, wolframite, rare zircon, apatite, and various sulfides [81]. Cassiterite is more abundant in the veins closer to the granites, occurring near the contact of the quartz lodes with the bedrock, whereas wolframite tends to increase away from the granites and occurring disseminated within the lodes [85].

The aplite-pegmatites exhibit granular hypidiomorphic textures and are composed of quartz, microcline, albite, muscovite, apatite, zircon, and rutile, with tourmaline present only in some [86]. Additionally, the Li-Sn-bearing aplite-pegmatites also have topaz, spodumene, lepidolite, cassiterite, columbite-tantalite, and Li phosphates of the amblygonite–montebrasite series [86].

The Ba-Pb-Zn quartz breccia veins occur along the ENE–WSW and NNE–SSW faults, with maximum thicknesses of 3 m and reaching 2500 m in length [85]. These quartz

breccias present a xenomorphic texture, with quartz and baryte as the predominant minerals within a mineralogical suite that also includes galena, sphalerite, muscovite, chlorite, and apatite [78]. Baryte constitutes 20%–30% of these mineralized structures and is present in two varieties: massive and saccaroid [78].

## 3. Materials and Methods

### 3.1. Heavy Mineral Sample Selection and Sample Preparation

Fifty-seven alluvial heavy mineral (HM) samples, distributed along a perpendicular strip to the Segura Massif contact, were selected (Figure 1) for HM analysis at Laboratório Nacional de Energia e Geologia (LNEG) facilities. Representativeness of the main out-cropping lithologies, with special attention to contributions from the different granite and mineralized bodies, was considered for sample selection, and further confirmed by the HM assemblages yielded through HM analyses. A Leica Wild M3X binocular microscope (Leica, Wetzlar, Germany) was used for mineral identification and hand-picking. A hand magnet able to attract magnetic and paramagnetic minerals (considered in this work as the mineral magnetic fraction), was also used for confirmation or screening of various mineral phases. Minerals were identified based on their physical properties (specific gravity, magnetism, color, luster, crystal shape, cleavage, and fluorescence under UV light), and TiO$_2$ polymorphs were hand-picked (850 rutile, 1125 anatase, and 295 brookite grains), mounted in epoxy, and polished for subsequent chemical analysis. It should be noted that field sampling, HM separation by specific gravities and magnetism, and partial HM studies were performed in previous national exploration projects led by LNEG and its precursor institutions. Magnetic and non-magnetic fractions produced then were subsequently made available for this study. Figure 2 summarizes the alluvial HM separation process and sample preparation.

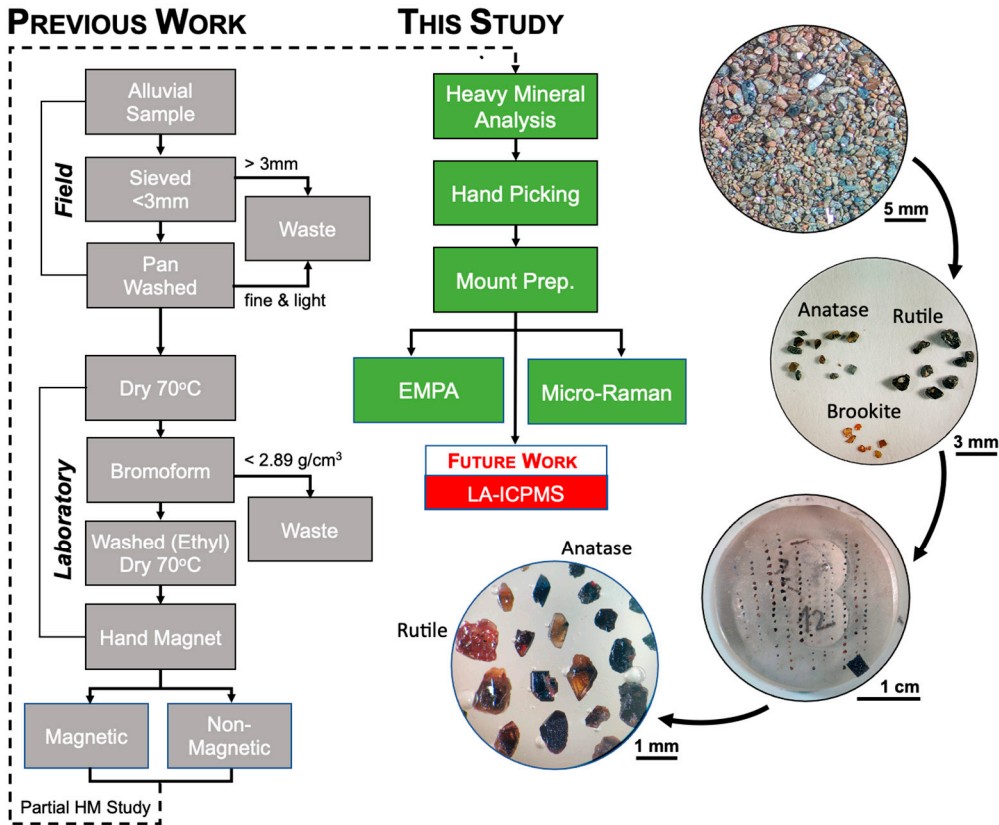

**Figure 2.** Heavy mineral processing scheme and sample preparation [37]. See text for details.

### 3.2. Heavy Mineral Analyses and Mineral Abundance Mapping

Mineral quantification was based on the interval scale adapted from [87], where intervals reflect abundance of each identified mineral in its corresponding mineral fraction (magnetic or non-magnetic). From lowest to highest abundance, this scale's intervals are: <1%; 1%–5%; 5%–25%; 25%–50%; 50%–75%; 75%–100%. The error associated with these measurements is approximately 5%. After mineral identification and semi-quantitative HM analysis, a statistical study of the mineral associations was performed by recombining the relative abundances of both magnetic and non-magnetic fractions. Average values of the relative abundance of $TiO_2$ polymorphs, and other key minerals (e.g., cassiterite, wolframite, and scheelite), were then projected onto mineral-distribution maps. Considering the type of data, mineral distribution maps were created with ArcGIS software (version 10.8.1, ESRI, Redlands, CA, USA) using the inverse distance weighted interpolation method (IDW). The IDW method estimates the values of an unknown area on the assumption that the points with known values have a greater influence on the calculated estimates for the areas closest to these points and decreases as the distance to the point increases. A similar approach was applied to the trace element compositional data subsequently acquired by EPMA for the three $TiO_2$ polymorphs.

### 3.3. Micro-Raman Spectroscopy

A HORIBA XPlora Raman spectrometer, equipped with a 785-nm near-infrared laser and coupled with an Olympus[TM] microscope, from the HERCULES Laboratory facility (University of Évora, Portugal), was used to record the Raman spectra of mounted grain of $TiO_2$ polymorphs. The system uses a thermo-electrically cooled charge-coupled device detector (CCD). At least three spectra per sample were acquired in the 100–2000 $cm^{-1}$ range. The 50x objective was used for all samples, and the measuring time, laser power, and number of accumulations were adjusted to avoid thermal damage and to obtain a good signal-to-noise ratio. The instrument itself was controlled using LabSpec software. The collected Raman spectra were further processed in GRAMS (ThermoFisher Scientific[TM]).

### 3.4. Electron Probe Micro-Analyses (EPMA)

Major and trace element composition of rutile (n = 979 analyses), anatase (n = 1128) and brookite (n = 105), back-scattered electron (BSE) images, and X-ray maps were obtained by electron probe micro-analyzer (EPMA) at Faculdade de Ciências da Universidade de Lisboa (Portugal), using a JEOL JXA 8200 instrument equipped with 4 wavelength-dispersive spectrometers (WDS), secondary and back-scattered electron detectors, and an energy-dispersive spectrometer (EDS). The analyses were performed using a 5-μm diameter beam, 25 nA of beam current, and 15 kV of accelerating voltage. Rutile (Ti), hematite (Fe), skutterudite (Co), cassiterite (Sn), metal tungsten (W), metal niobium (Nb), $LiTaO_3$ (Ta), Cr-oxide (Cr), metal V (V), zirconia (Zr), apatite (P), jadeite (Na), stibnite (S), benitoite (Ba), Bi-selenide (Bi), periclase (Mg), galena (Pb), Ga-arsenide (As), sanidine (K), diopside (Si, Ca), plagioclase (Al), willemite (Zn), Ni-silicide (Ni), rhodonite (Mn), cuprite (Cu), Ag metal, and Au metal were used as standards. Standard ZAF corrections were applied.

## 4. Results

### 4.1. Alluvial Heavy Mineral Analysis

Heavy mineral analysis, consisting of the identification and quantification of heavy minerals, was performed in fifty-seven alluvial samples from Segura, in both magnetic and non-magnetic concentrate fractions. Mineral identification was based on their physical properties (specific gravity, magnetism, color, habit/crystal shape, cleavage, luster, and fluorescence under UV light) under a binocular microscope. Table 1 summarizes the minerals identified according to their average abundance considering all 57 alluvial samples. It should be noted that the minerals that could not be identified were lumped together as "undifferentiated" minerals (Und. minerals).

**Table 1.** Heavy mineral species identified in the Segura alluvial samples. Mineral percentage (%) corresponds to the average abundance of each mineral in the total of the 57 HM concentrates (magnetic + non-magnetic) under study.

| Mineral | % | Mineral | % | Mineral | % |
|---|---|---|---|---|---|
| Iron oxyhydroxide | 43.18 | Andalusite | 0.78 | Corundum | 0.03 |
| Tourmaline | 11.52 | Biotite | 0.74 | Classic monazite | 0.03 |
| Magnetite | 11.03 | Brookite | 0.72 | Epidote | 0.01 |
| Ilmenite | 5.61 | Zoisite | 0.42 | Chlorite | 0.01 |
| Cassiterite | 4.57 | Zircon | 0.41 | Sillimanite | 0.01 |
| Nodular monazite | 3.41 | Scheelite | 0.38 | Xenotime | 0.01 |
| Anatase | 2.73 | Altered pyrite | 0.34 | Topaz | <0.01 |
| Wolframite | 2.20 | Garnet | 0.30 | Magnetic leucoxene | <0.01 |
| Baryte | 2.19 | Gold | 0.12 | Nonmagnetic leucoxene | <0.01 |
| Apatite | 2.03 | Muscovite | 0.07 | | |
| Rutile | 1.18 | Cinnabar | 0.05 | Und. minerals | 6.33 |
| Total | | | | | 100% |

The iron oxyhydroxide, with the highest average abundance, stood out, reflecting its significant amount in all the studied samples, whereas the other abundant minerals are restricted to specific samples (see the mineral association in Section 4.1.1 for details).

Regarding the $TiO_2$ polymorphs, anatase is the most abundant (2.73%), followed by rutile (1.18%) and brookite (0.42%). A map showing the relative abundance of $TiO_2$ minerals in each sample is presented in Figure 3, where the predominance of anatase in the metasediments and of rutile in granites stands out. The relationship between the relative abundances of the $TiO_2$ minerals and their genesis is discussed in Section 5.1.3. Considering other minerals of interest for this study, i.e., Sn and W minerals, cassiterite is the most abundant (4.57%), followed by wolframite (2.20%), while scheelite is much scarcer (0.34%), which mirrors the weighted abundance and regional diversity of the Sn-W ore systems (Figure 3). Tourmaline is quite abundant, reflecting both igneous and metamorphic tourmaline-bearing rocks. The average abundance of baryte is also noteworthy, expressing the profusion of Alpine Ba-Pb mineralized structures in the area.

The $TiO_2$ polymorphs were characterized and identified based on their main physical properties (Figure 4), a task that is not always straightforward given the typical complex nature of alluvial samples and consequent heterogeneity of the minerals' features.

Rutile was identified in the non-magnetic fraction of 54 of the alluvial samples studied. A wide variety of rutile colors were observed, including black, reddish-black, grayish-black, blood-red, yellow, or brown crystals; in most cases, they are sub-translucent, with an adamantine or metallic luster. Most of the rutile crystals are anhedral, subrounded with grain sizes ranging from fine (<0.5 mm) to coarse (>1 mm). In samples with appreciable amounts of rutile and cassiterite, it was difficult to distinguish them, particularly in the case of black crystals with anhedral shapes. The cassiterite "tinning test" technique, adapted from [87], was used to screen out the cassiterite grains. Prismatic rutile crystals, with bipyramidal terminations and vertical striations on the prism faces, often showing evidence of abrasion, were identified in 21 samples. "Elbow" twins were also observed in some crystals, as well as thinner crystals (<0.1–0.2 mm), with slender acicular shapes, as described in [88].

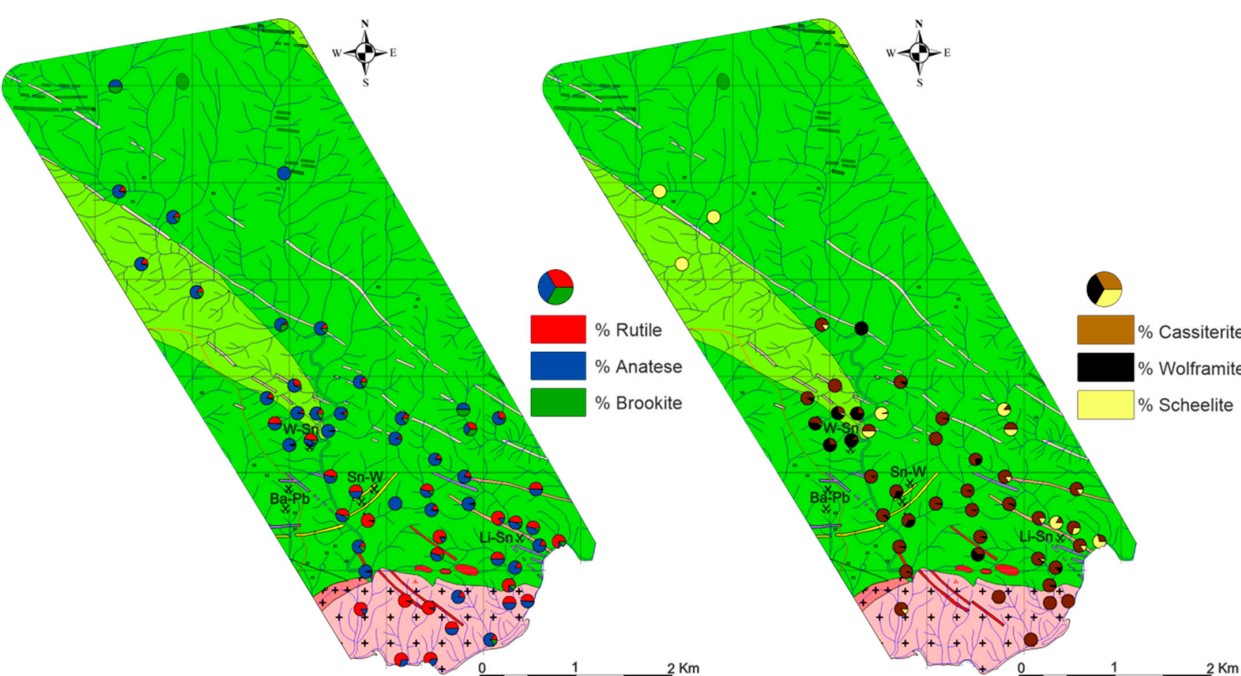

**Figure 3.** Relative abundance of TiO₂ polymorphs and Sn and W ore minerals for each alluvial sample studied, modified from [37]. See text for details. Geological background as in Figure 1c.

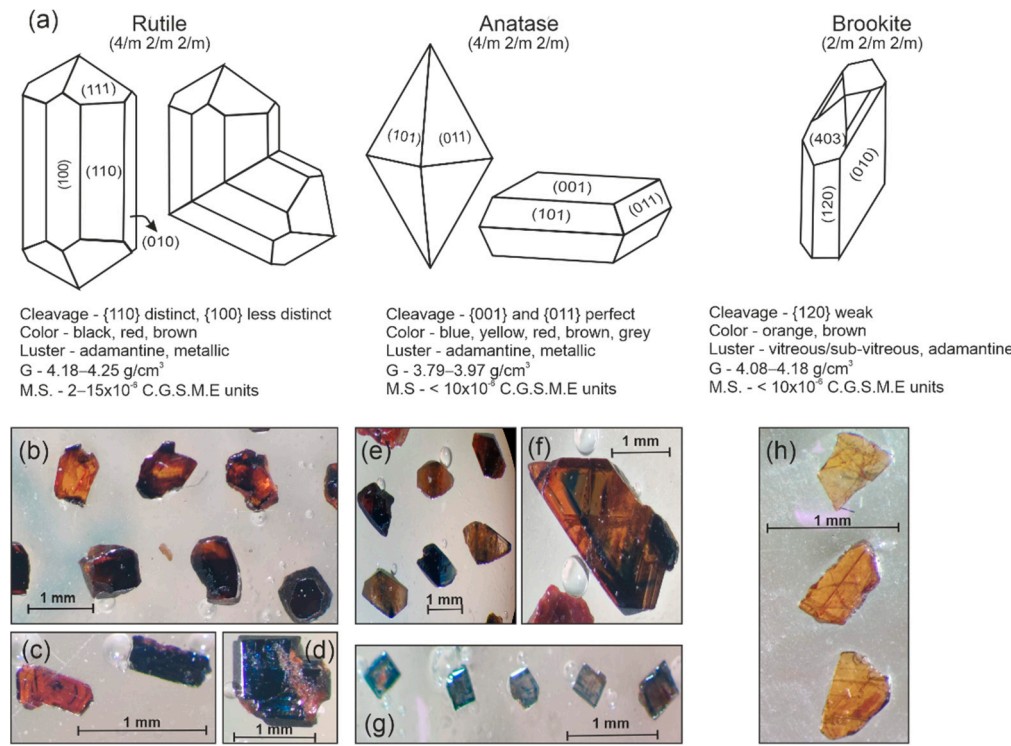

**Figure 4.** Common crystal shapes of TiO₂ polymorphs (adapted from [89]) and main physical properties (**a**): cleavage, color, luster, and G (specific gravity) [88], magnetic susceptibility (M.S.) [87]. Hand-picked grains of rutile (**b**–**d**), anatase (**e**–**g**), and brookite (**h**). Rutile prismatic crystals with bipyramidal termination (**c**). Rutile "elbow" twin (**d**). Bipyramidal and color-zoned anatase (**e**,**f**). Tabular blue anatase (**g**).

Anatase was identified in all 57 samples and stood out from the other heavy minerals due to its characteristic tetragonal {011} bipyramidal shape (Figure 4). However, due to the prominent development of the {001} parallelohedron truncating the bipyramid, basal forms are also present. Anatase presents extensive chromatic variability, with brown, yellow, red, blue, and grayish colors, and a heterogeneous granularity, ranging from fine (<0.5 mm) to coarse (>1 mm) grained. Anatase grains exhibit adamantine or metallic luster and are more transparent when weakly colored and opaquer when strongly colored. Because anatase and rutile crystallize in the same crystallographic system, some prismatic crystals with bipyramidal terminations are difficult to distinguish, as noted elsewhere [88]. Thus, doubtful grains were screened by Raman spectroscopy, to ensure the correct $TiO_2$ mineral phase identification, prior to chemical analysis.

Brookite, the rarest of the three $TiO_2$ polymorphs, was identified in 42 alluvial samples, with a relative abundance greater than 1% in only 4 of them. Brookite was fairly easy to identify based on its physical and optical properties (Figure 4). Tabular {010} shapes predominate, exhibiting typical [001] striation, as described in [88]. Crystals are mostly fine grained (<0.5mm) but may reach coarser dimensions (>1 mm). Brookite is transparent or translucent, with orange colors (sometimes with browner or yellowish hues) and a vitreous to sub-vitreous, sometimes adamantine, luster.

### 4.1.1. Heavy Mineral Assemblages

HM analysis revealed mineral assemblages that echo the bedrock lithologies of the sampling sites, indicating a low mineral dispersion, and thus enhancing the assessment of mineralizing footprints and vectors. Table 2 and Figure 5 show distinct and representative alluvial mineral associations for the samples collected within the Segura granites, in the SGC metasediments, and in the vicinity of the mineralized bodies.

In samples collected within the granites, heavy mineral assemblages are dominated by tourmaline, biotite, apatite, and garnet, with rutile (4.61%) > anatase (1.84%). Iron oxyhydroxides (Fe O-OH) dominate samples from the SGC metasediments, including samples close to the mineralized quartz lodes (Sn-W, W-Sn, and Ba-Pb), all with anatase (0.29%–6.61%) > rutile (0.14%–5.32%). Brookite is rare but, when present, is always below 1.07%.

**Table 2.** Heavy-mineral-analyses summary for representative samples collected within the SM granites, SGC metasediments, and in proximity to the distinct mineralization types in the study area.

| Granites | | SGC Metasediments | | Sn-W | | W-Sn | | Li-Sn | | Ba-Pb | |
|---|---|---|---|---|---|---|---|---|---|---|---|
| **Mineral** | **%** | **Mineral** | **%** | **Mineral** | **%** | **Mineral** | **%** | **Mineral** | **%** | **Mineral** | **%** |
| Tourmaline | 57.71 | Fe O-OH. | 52.93 | Fe O-OH. | 42.60 | Fe O-OH. | 37.24 | Ilmenite | 49.92 | Fe O-OH. | 35.50 |
| Biotite | 13.86 | Magnetite | 31.75 | Cassiterite | 35.50 | Wolframite | 37.24 | Magnetite | 19.97 | Baryte | 22.19 |
| Apatite | 7.69 | Anatase | 6.61 | Wolframite | 17.04 | Cassiterite | 10.63 | Cassiterite | 12.48 | Magnetite | 14.20 |
| Rutile | 4.61 | Altered Pyrite | 2.54 | Apatite | 1.70 | Anatase | 4.25 | Scheelite | 4.99 | Anatase | 5.32 |
| Altered pyrite | 2.77 | Ilmenite | 2.54 | Epidote | 0.56 | Und. minerals | 4.25 | Altered pyrite | 3.99 | Cassiterite | 5.32 |
| Garnet | 2.77 | Und. minerals | 1.59 | Nod. Monazite | 0.56 | Magnetite | 2.97 | Tourmaline | 3.99 | Rutile | 5.32 |
| Fe Ox-hyd. | 2.77 | Brookite | 0.31 | Anatase | 0.29 | Brookite | 0.86 | Anatase | 1.00 | Ilmenite | 2.84 |
| Zoisite | 2.77 | Gold | 0.06 | Brookite | 0.29 | Scheelite | 0.86 | Rutile | 1.00 | Tourmaline | 2.84 |
| Anatase | 1.84 | Scheelite | 0.06 | Gold | 0.29 | Ilmenite | 0.50 | Zircon | 1.00 | Zoisite | 2.84 |
| Zircon | 1.84 | | | Rutile | 0.29 | Tourmaline | 0.50 | Nod. Monazite | 0.67 | Andalusite | 1.07 |
| Ilmenite | 0.46 | | | Topaz | 0.29 | Baryte | 0.14 | Zoisite | 0.67 | Brookite | 1.07 |
| Magnetite | 0.46 | | | Scheelite | 0.29 | Gold | 0.14 | Brookite | 0.17 | Altered pyrite | 0.47 |
| Andalusite | 0.37 | | | Und. minerals | 0.29 | Muscovite | 0.14 | Gold | 0.17 | Wolframite | 0.47 |
| Muscovite | 0.06 | | | | | Rutile | 0.14 | | | Gold | 0.18 |
| | | | | | | Zircon | 0.14 | | | Scheelite | 0.18 |
| | | | | | | | | | | Zircon | 0.18 |

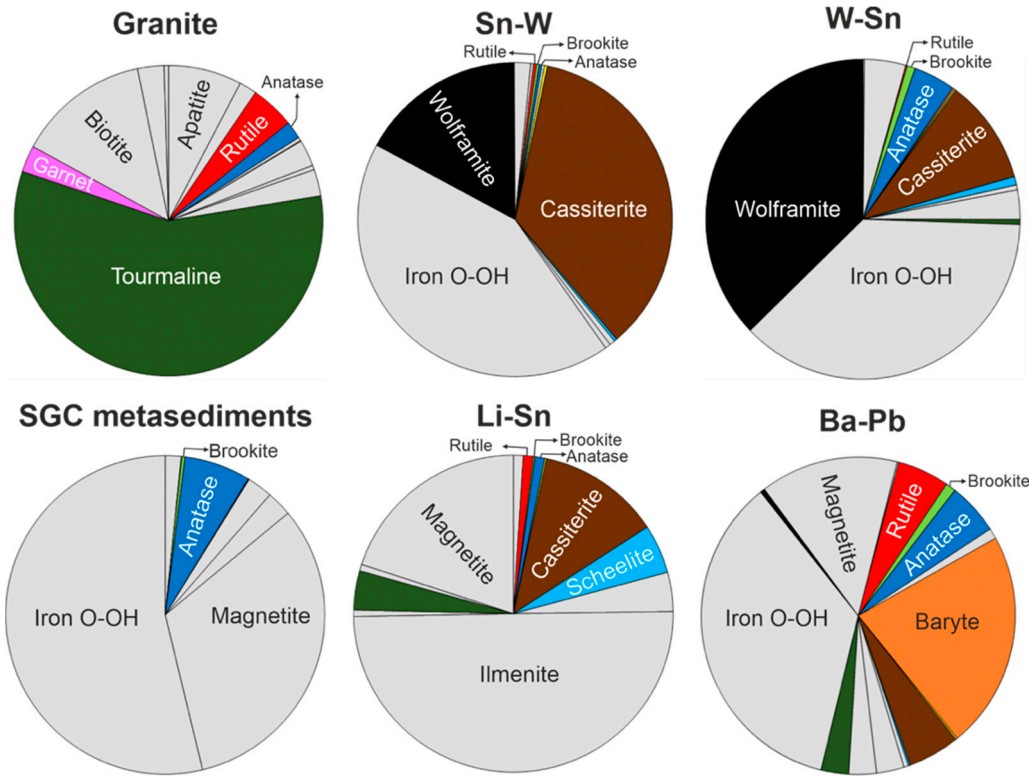

**Figure 5.** Alluvial heavy mineral assemblages with relative mineral abundance representative of the main Segura bedrock lithologies, including mineralized components. Adapted from [37].

Notably, an abundance of magnetite in almost all the metasedimentary samples was observed, except those associated with the W-Sn and Sn-W veins, where magnetite is minor or absent. Samples proximal to Sn-W mineralized veins contain more cassiterite than wolframite, while in those proximal to W-Sn veins the exact opposite occurs. Around the Li-Sn aplite-pegmatite dykes, alluvial samples are also cassiterite-rich. Another ore mineral, scheelite, although scarce, seemed to be ubiquitous and particularly abundant near the W-Sn veins and around the Li-Sn dykes. Baryte is obviously abundant in the vicinity of the Ba-Pb veins. The variability observed in some of the samples, particularly the baryte-rich ones, reflects the complexity of the area, with several mafic and felsic igneous rocks and Sn-bearing aplite dykes.

4.1.2. Alluvial Heavy Mineral Abundance Maps

The relative abundance of $TiO_2$ polymorphs and other key minerals (cassiterite, wolframite, and scheelite) was used to create inverse distance weighted (IDW) interpolated mineral-distribution maps (Figure 6).

Rutile shows well defined positive anomalies (Figure 6a), pinpointing the Segura Massif granites and main Sn and W mineralized structures. Anatase shows higher abundance, with a more widespread distribution and weaker positive anomalies (Figure 6b). However, marked anatase negative anomalies define a contact halo, within the SGC metasediments, around the Segura Massif, as well as a NW–SE trend aligned with the major Sn-W and W-Sn mineralized quartz veins. Brookite abundance is generally low (Figure 6c), except in the central eastern part of the map, where a few higher values delineate a large positive anomaly, potentially related to the intrusive tonalite bodies that are present to the NE of the study area.

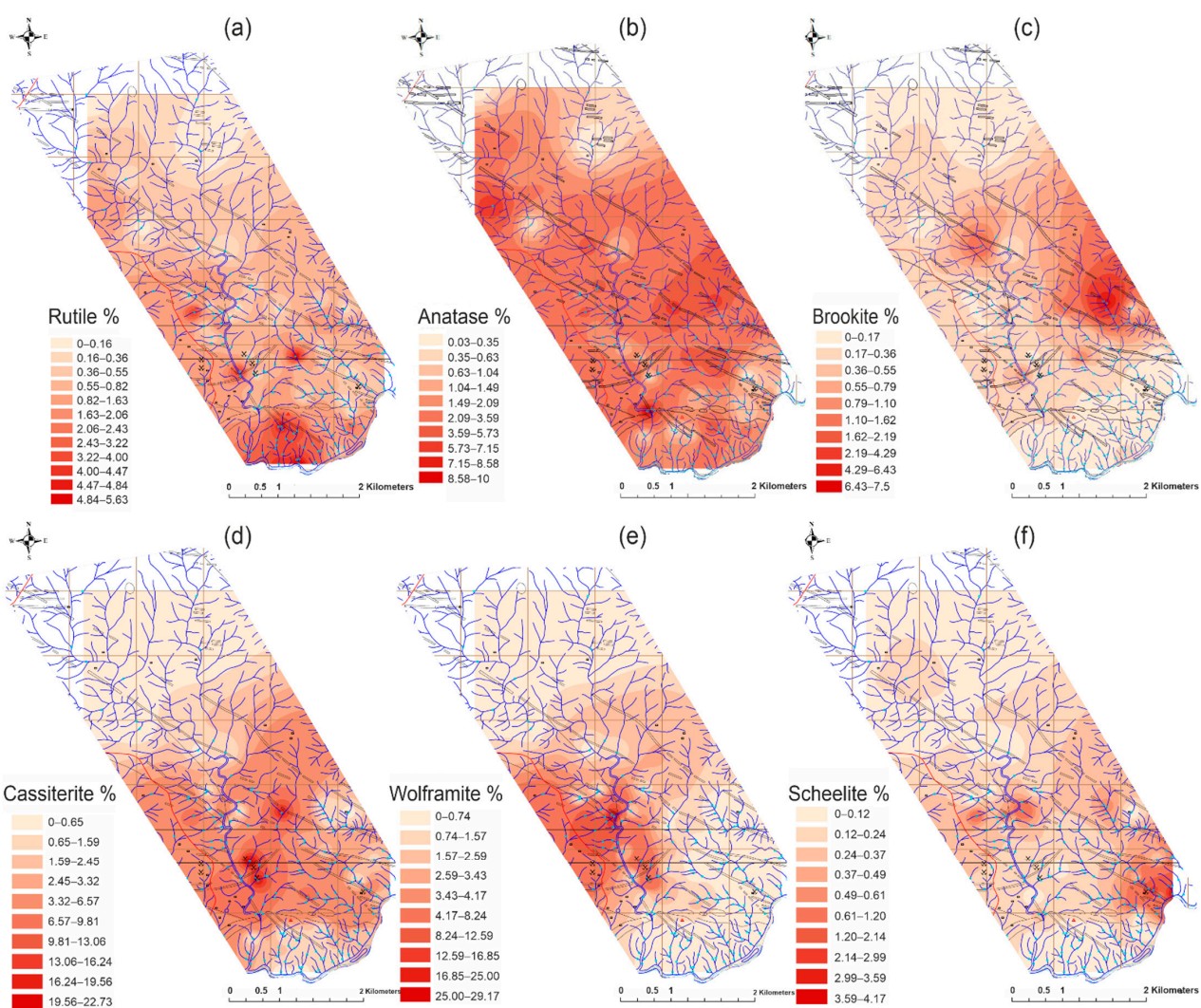

**Figure 6.** Mineral relative abundance distribution maps for alluvial rutile (**a**), anatase (**b**), brookite (**c**), cassiterite (**d**), wolframite (**e**), and scheelite (**f**). Inverse distance weighted interpolated maps created with ArcGIS software [37].

The abundance distribution maps for the ore minerals (cassiterite, wolframite, and scheelite—Figure 6d, 6e and 6f, respectively) show a common depletion within the Segura Massif; however, their mismatch outside of the granites reflects the diversity and complexity of the tin and tungsten systems. Cassiterite shows high background values, in a 2 km strip within SGC metasediments, and is most abundant in the alluvial samples with a meaningful contribution from the Sn-W mineralized zones. Marked positive anomalies match the main Segura mine tailings. Wolframite does not occur in as many samples as cassiterite. Since it is a friable mineral, the highest concentration of wolframite should be proximal to its source. In addition to the anomaly that is paired with that of cassiterite, correlated with the Segura mine Sn-W veins, a larger positive wolframite anomaly can be seen to the NW, documenting W-dominated quartz veins. Note that scheelite has two positive anomalies in the same area, as well as a wider anomaly, in the SE sector of the map, close to the Li-Sn mineralized aplite-pegmatites.

These results support previous mineral exploration studies [74,90] and add new relevant facts. The HM analysis showed the immense potential of alluvial mineral assemblages as proxies for local geology and to the metamorphic/metasomatic processes related to productive intrusive bodies in similar terrains, where dispersion factors, such as geomor-

phology and hydraulic sorting, among others, prevented significant transport and did not significantly influence mineral dispersion.

### 4.2. Trace Elements in TiO₂ Minerals—EPMA Data

Polished mounts of hand-picked rutile, anatase, and brookite were prepared for chemical analysis. The correct identification of some rutile and anatase grains with doubtful morphological characteristics was certified by Raman spectroscopy (Figure 7a) prior to electron probe micro analysis (EPMA). Each grain was also inspected for heterogeneities and inclusions with backscattered electron (BSE) imaging (Figure 7b–i).

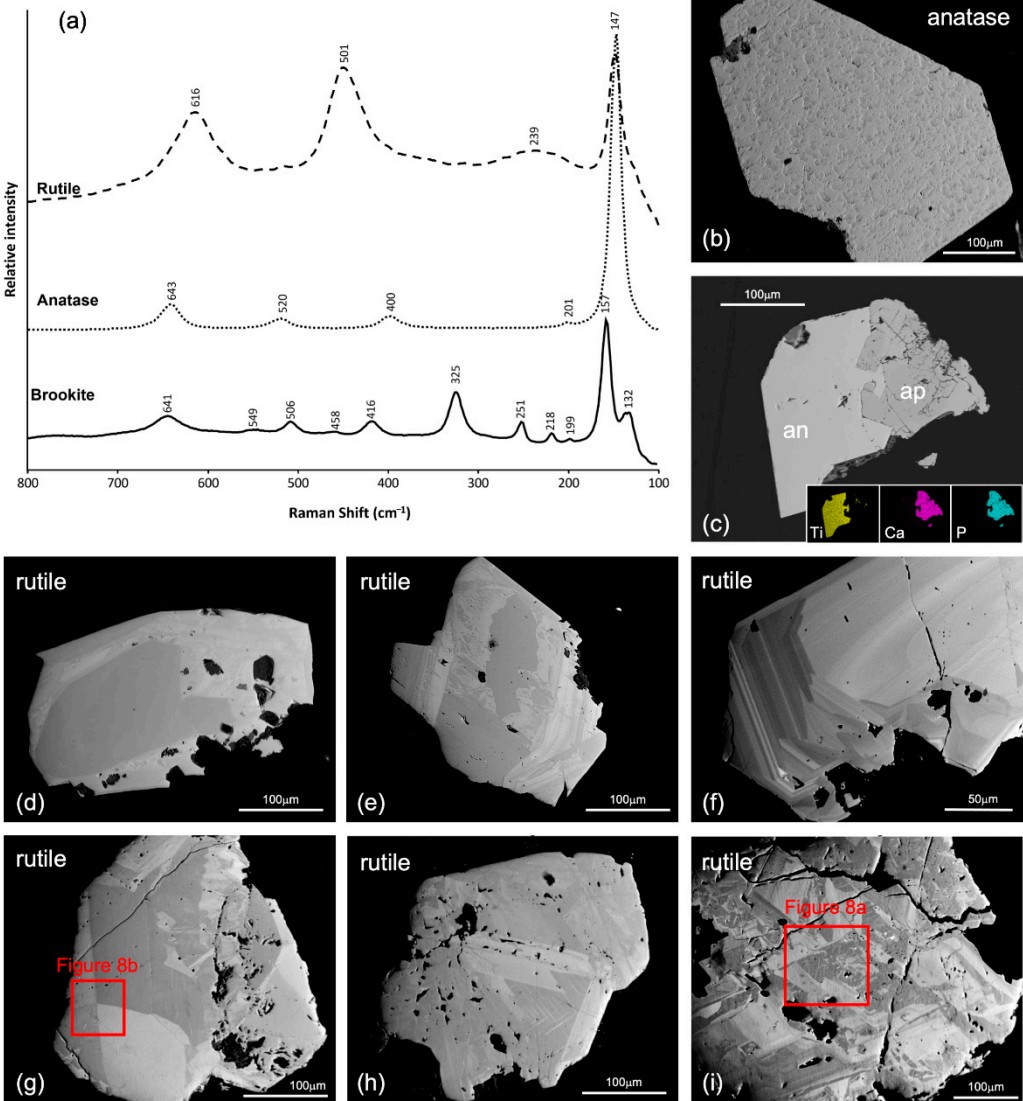

**Figure 7.** Raman spectra of Segura alluvial TiO₂ polymorphs to certify correct identification prior to EPMA (**a**). Backscattered electron (BSE) EPMA images showing compositional homogeneity for anatase (**b**,**c**) and some aspects of rutile compositional intra-granular variability (**d**–**i**): homogeneous anatase (**b**); anatase (an) intergrown with apatite (ap) with X-ray maps for Ti, Ca, and P included in the bottom right (**c**); zoned rutile with homogeneous trace element-poor darker cores and heterogeneous trace element-rich lighter rims (**d**–**e**); oscillatory zoning (**d**–**f**); sector zoning (**f**–**i**); and more irregular, complex, and patchy zoning patterns (**h**,**i**).

Anatase and brookite are typically compositionally homogenous and devoid of inclusions. However, some rutile crystals exhibit well-developed compositional zoning patterns

(oscillatory zoning, sector zoning, patchy and irregular zoning), which are very clear in BSE imaging (Figure 7). The observed zoning, reflecting differences in the average atomic number, reproduces heterogeneous distributions of trace elements, which were evaluated by X-ray mapping (Figure 8). The lighter zones are enriched in W, Nb, Ta, Fe, Sn, and V, contrasting with the darker zones, which are generally trace element poor (Figure 8); the latter mostly occurring in grain cores (Figure 7d,e). The sector zoning (i.e., Figure 7g) sometimes shows trace element contents controlled by crystal forms and faces, with Sn and V paired with Ti-rich sectors, and W, Nb, Ta, and Fe enriched in Ti-poor sectors (Figure 8b).

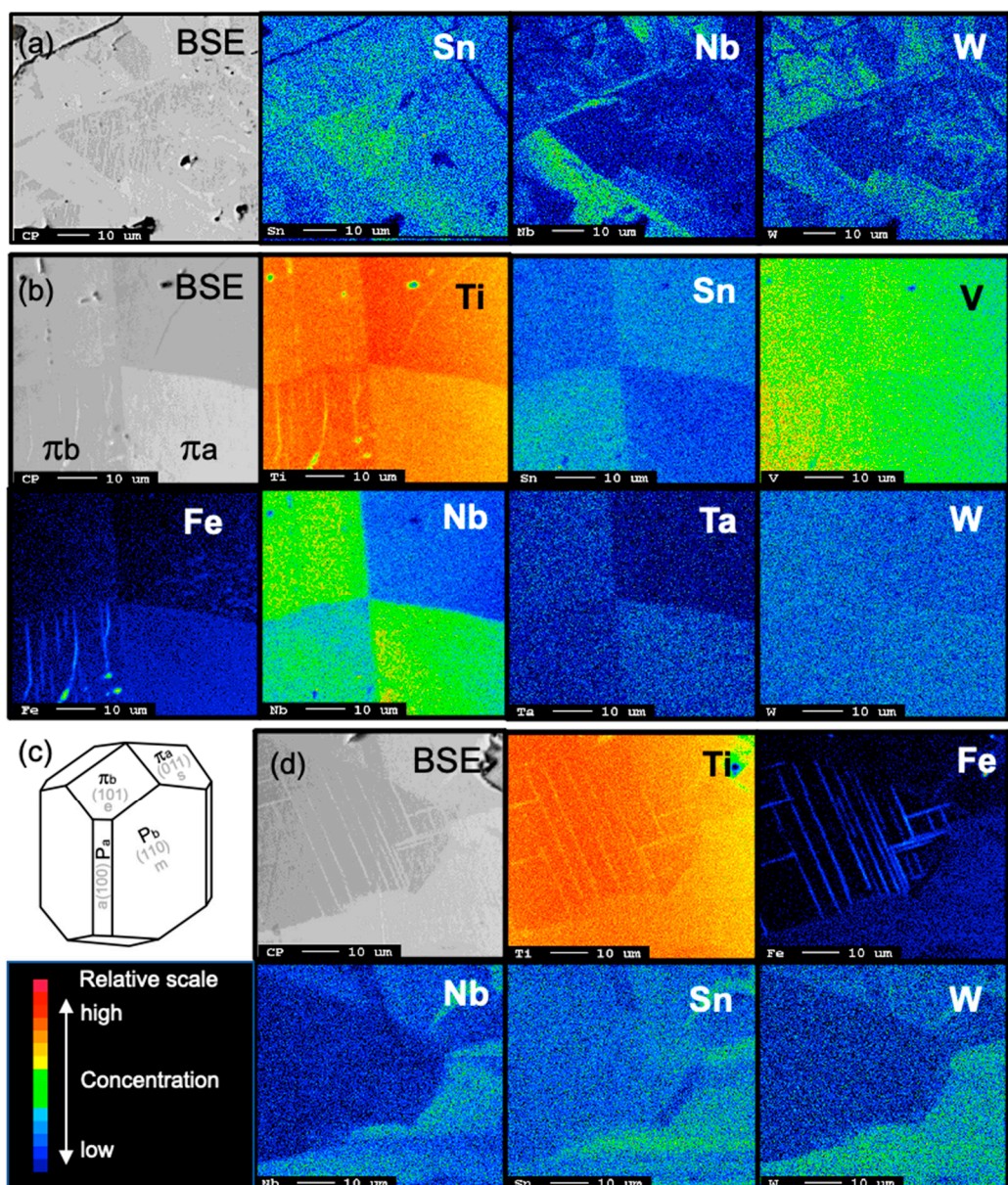

**Figure 8.** Backscattered electron (BSE) images and selected compositional X-ray maps (EPMA) from three rutile grains (**a**,**b**,**d**) showing variable trace-element contents. Crystallographic forms of tetragonal rutile—{100} prism (Pa faces); {110} prism (Pb faces); {101} bipyramid (*πa and πb* faces) from [70] (**c**). Note the marked crystallographic control on trace element composition of rutile (**b**) by alternating high Ti, Sn, V bipyramidal faces (*πa*, for example) with high W, Nb, Ta, and Fe faces (*πb*).

Table 3 presents a summary of the average EPMA data for the most significant trace elements in rutile (n = 979 analysis), anatase (n = 1128) and brookite (n = 105) from the studied alluvial samples, including V, Cr, Fe, Sn, Nb, Ta, W, and Zr. A representative dataset

is attached to the paper as Supplementary Material (see Table S1), while the full dataset is available in [37].

**Table 3.** Trace element minimum, maximum, average and median concentration values (ppm) for Segura alluvial rutile, anatase and brookite.

| Rutile (ppm) | V | Cr | Fe | Sn | Nb | Ta | W | Zr |
|---|---|---|---|---|---|---|---|---|
| Minimum | <b.d.l | <b.d.l | <b.d.l | <b.d.l | <b.d.l | <b.d.l | <b.d.l | <b.d.l |
| Maximum | 10,937 | 4201 | 29,258 | 55,665 | 85,347 | 19,139 | 57,173 | 1577 |
| Average | 1712 | 168 | 5882 | 4462 | 5723 | 890 | 5126 | 68 |
| Median | 1271 | <b.d.l | 4508 | 882 | 3845 | <b.d.l | 1491 | <b.d.l |
| Std. Dev. | 1388 | 370 | 5196 | 7900 | 6269 | 1990 | 8898 | 163 |
| **Anatase (ppm)** | | | | | | | | |
| Minimum | <b.d.l | <b.d.l | <b.d.l | <b.d.l | <b.d.l | <b.d.l | <b.d.l | <b.d.l |
| Maximum | 5289 | 1006 | 17,699 | 4892 | 12,296 | 16,895 | 10,523 | 992 |
| Average | 754 | 23 | 472 | 85 | 1575 | 261 | 694 | 99 |
| Median | 877 | <b.d.l | 894 | <b.d.l | 1139 | <b.d.l | <b.d.l | <b.d.l |
| Std. Dev. | 545 | 88 | 280 | 387 | 1510 | 833 | 1566 | 193 |
| **Brookite (ppm)** | | | | | | | | |
| Minimum | <b.d.l | <b.d.l | <b.d.l | <b.d.l | <b.d.l | <b.d.l | <b.d.l | <b.d.l |
| Maximum | 2855 | 1526 | 9553 | 6333 | 4649 | 2334 | 3457 | 429 |
| Average | 762 | 141 | 2268 | 123 | 1082 | 401 | 373 | 42 |
| Median | 714 | <b.d.l | 1780 | <b.d.l | 811 | <b.d.l | <b.d.l | <b.d.l |
| Std. Dev. | 622 | 236 | 1748 | 640 | 862 | 575 | 633 | 115 |

(b.d.l.)—below detection limit.

Rutile shows higher trace element contents relative to anatase and brookite (Table 3, Figure 9), with significant median values for Nb (3845 ppm), Fe (4508 ppm), W (1491 ppm), V (1271 ppm), and Sn (882 ppm), reaching maximum values of 85,347 ppm Nb, 57,173 ppm W, 55,665 ppm Sn, 29,258 ppm Fe, and 10,937 ppm V in samples within the granites. Medians for Ta, Zr and Cr in the rutile are below the detection limit (b.d.l.); however, Ta can reach 19,139 ppm, while Cr and Zr do not exceed 4200 ppm and 1577 ppm, respectively.

In anatase only Nb (1139 ppm), Fe (894 ppm) and V (877 ppm) medians are above detection, but the maximum concentrations of Ta (16,895 ppm) are greater than those of Nb (12,296 ppm), as opposed to rutile. It is also important to highlight the maximum concentrations of Fe, W, V and Sn in anatase, which can reach 17,699 ppm Fe, 10,523 ppm W, 5289 ppm V and 4892 ppm Sn. Regarding brookite, only Fe (1780 ppm), Nb (811 ppm), and V (714) show considerable median concentrations. Note also that the median Fe content in brookite is higher than in anatase, and that this element clearly stands out as the main trace in brookite.

Considering the characteristic mineral assemblages defined in the HM study (Table 2, Figure 5), our dataset recognizes distinct trace element compositional features in the $TiO_2$ polymorphs (Figure 9). Rutile grains representing the Segura granites' HM component show the highest trace elements enrichment, especially in V, Sn, Fe, Nb and W with concentrations that can exceed 10,000 ppm (Figure 9a). In samples representing the CXG metasediments (Figure 9b), the trace element content of $TiO_2$ polymorphs is usually lower, except for some W-rich anatase crystals, whose compositions may have been inherited from precursor minerals or may reveal interaction with W-rich fluids. The trace element contents of the $TiO_2$ polymorphs from samples near the mineralized bodies are heterogeneous, varying between 100 and 10,000 ppm (Figure 9c–f), reflecting the multiplicity of sources and the diversity of mineral-forming processes. Despite this heterogeneity, a great number of high values are recorded in these samples. The Sn-enriched signature of the $TiO_2$ minerals associated with the W-Sn veins is noteworthy, contrasting with their generally Sn-depleted signature associated with the Sn-W veins. Furthermore, in many samples, the W and Sn contents correlate negatively. In addition, the samples associated with the Ba-Pb veins are the only in which Sn in anatase is higher than Sn in rutile.

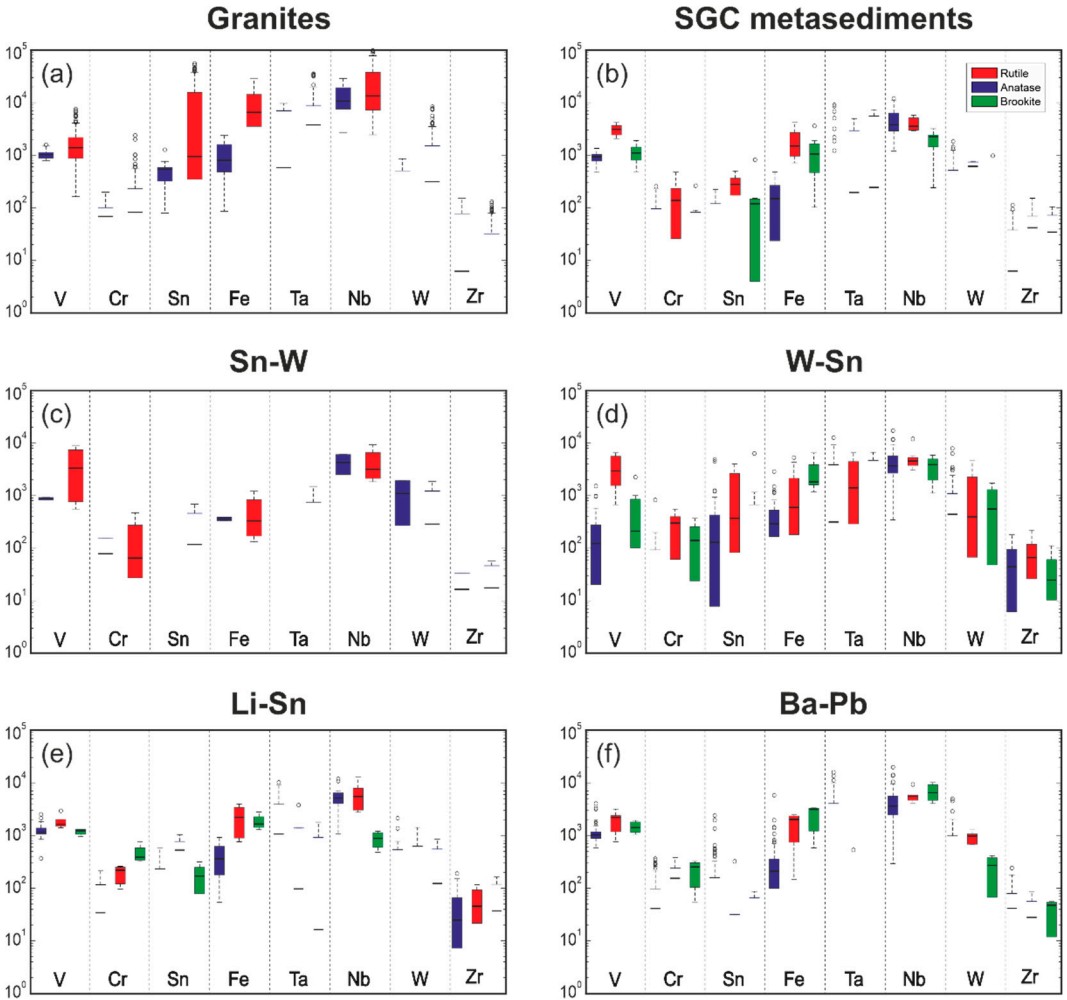

**Figure 9.** Boxplots of EPMA data for noteworthy trace elements in TiO₂ polymorphs [37]. Each boxplot includes all analyzed grains from samples representing the six Segura mineral assemblages defined by HM analysis. Granites (**a**); SGC metasediments (**b**); Sn-W (**c**); W-Sn (**d**); Li-Sn (**e**); Ba-Pb (**f**).

## 5. Discussion

### 5.1. Controls on TiO₂ Polymorphs' Trace Element Composition

In natural TiO₂ minerals, Ti⁴⁺ can be replaced by a wide variety of other hexa-, penta-, tetra-, tri-, di-, and monovalent cations [56,58,91–93]. The incorporation of trace elements into minerals is closely related to the geochemical affinity between elements, which is largely controlled by their effective ionic radii and charge, to the intrinsic mineral factors associated with their crystal structures, and to the physical-chemical parameters characteristic of the forming environments.

#### 5.1.1. Structure of TiO₂ Polymorphs

In nature, TiO₂ occurs mostly in three distinct structural configurations, corresponding to the three polymorphs, rutile, anatase, and brookite [62]. Ti⁴⁺ cations are coordinated by six O²⁻ anions in all three crystalline structures, forming distorted TiO₆ octahedrons, with distinct arrangements (sharing vertices or edges) and degrees of distortion (Figure 10). Rutile and anatase both crystallize in the tetragonal system. In rutile, each octahedron shares only two edges with neighboring octahedrons, forming [001] rows, whereas in anatase, (001) octahedron layers overlap by sharing four edges. In brookite, each octahedron shares three edges with its neighbors, creating an orthorhombic structure. The sharing of additional edges in anatase and brookite, when compared to rutile, makes their structure more rigid

and less elastic, limiting their capacity to accommodate the distortion resulting from the incorporation of trace elements. In part, this explains the observed differences in the type and extension of trace element substitution between the $TiO_2$ polymorphs in this study (Table 3, Figure 11).

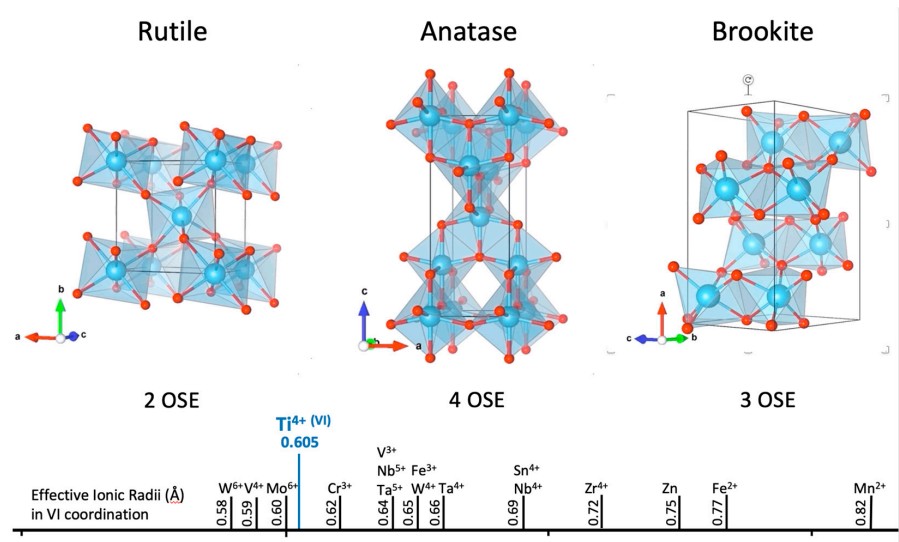

**Figure 10.** Distinct arrangement of $TiO_6$ octahedron (sharing vertices and edges) in $TiO_2$ polymorph structures. Note the 2 characteristic octahedron sharing edges (OSEs) in rutile, 4 OSEs in anatase, and 3 OSEs in brookite, a factor that controls trace-element contents (see text for detailed explanation). The 6-fold coordination effective ionic radii of $Ti^{4+}$ and other trace elements [94] relevant to these Ti minerals are also provided. $TiO_2$ polymorph structures were produced with VESTA software [95] and can be seen in 3D animated gifs (Figures S1–S3).

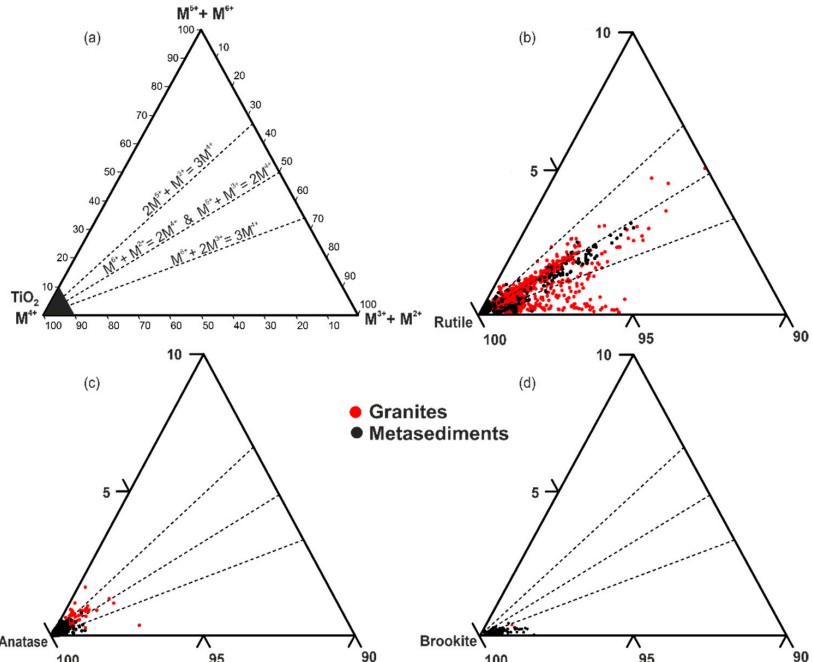

**Figure 11.** Ternary plot ($M^{4+} - M^{2+} + M^{3+} - M^{6+} + M^{5+}$) for $TiO_2$ polymorph composition showing the main heterovalent cation-substitution vectors (Equations (2)–(5)) (**a**). Black triangle represents the zoom area of figures (**b**–**d**). Segura alluvial rutile composition (**b**). Segura alluvial anatase composition (**c**). Segura alluvial brookite composition (**d**). Adapted from [37].

Trace-element substitution into minerals can also be a function of minerals' crystal forms and their subsequent adjustment during crystal growth, as described for rutile in Section 4.2 and elsewhere [64,70]. The sector zoning observed in some bipyramidal rutile, displaying alternated Sn- and V-enriched faces and W-, Nb-, Ta- and Fe-enriched faces (Figure 8b), is likely to be a mean of minimizing structural distortions, resembling, on a different scale, the nearest-neighbor Al/Al avoidance principle [96], the main driving force behind Al/Si ordering in tetrahedral aluminosilicate structures [97]. The enrichment in Sn and V on the less $Ti^{4+}$-depleted faces suggests that both elements are incorporated into rutile via similar isovalent substitutions, implying that V should be present in its oxidized form ($V^{4+}$). The enrichment in the heterovalent cations ($W^{6+}$, $Nb^{5+}$, $Ta^{5+}$, and $Fe^{3+}$ or $Fe^{2+}$), observed in the overtly Ti-depleted bipyramidal faces, must instead follow paired substitutions to maintain charge balance.

### 5.1.2. Trace Element Substitution Mechanisms in $TiO_2$ Minerals

The significant substituting trace elements into rutile, anatase, and brookite lattices are those with effective ionic radii similar to $Ti^{4+}$ in octahedral coordination (Figure 10). The elements with 4+ ionic charge ($M^{4+}$), such as $Sn^{4+}$, $V^{4+}$, $Zr^{4+}$, $Nb^{4+}$ or $Ta^{4+}$, can enter the structure by the isovalent substitution of $Ti^{4+}$ (Equation (1)). For charges other than 4+, it is necessary to consider paired substitution mechanisms. This is the case for heterovalent highly charged ($M^{6+}$ and $M^{5+}$) and low charged ($M^{2+}$, $M^{3+}$) ions that substituted for $Ti^{4+}$ (Equations (2)–(6)).

$$M^{4+} = Ti^{4+} \tag{1}$$

$$M^{6+} + M^{2+} = 2Ti^{4+} \tag{2}$$

$$M^{6+} + 2M^{3+} = 3Ti^{4+} \tag{3}$$

$$2M^{5+} + M^{2+} = 3Ti^{4+} \tag{4}$$

$$M^{5+} + M^{3+} = 2Ti^{4+} \tag{5}$$

$$2M^{2+} = Ti^{4+} + [\ ] \tag{6}$$

$M^{2+}$ cations, such as $Fe^{2+}$, $Mn^{2+}$ or $Zn^{2+}$, can enter the structure according to Equations (2), (4), and (6). $M^{3+}$ cations, mostly $Fe^{3+}$, $V^{3+}$ or $Cr^{3+}$, but also Sc, Al and As, follow Equations (3) and (5). The $M^{5+}$ cations are typically represented by Nb, Ta, and Sb; the most common $M^{6+}$ cations are W and Mo; U is a rarer cation [56]. When $M^{5+}$ or $M^{6+}$ cations are incorporated, the excess positive charges can still be balanced by the presence of vacancies in the $Ti^{4+}$ position [55].

Although the substitution mechanisms are similar in all three polymorphs, their magnitude depends mainly on the total amount of defects and lattice distortions allowed for each $TiO_2$ polymorph structure, as discussed in the previous section.

In Segura, the $TiO_2$ polymorphs exhibit more than one trace element substitution mechanism, not exceeding 2%–3% in total for brookite and anatase but reaching almost 10% in rutile (Figure 11). The more widespread trace element substitution in rutile from the samples collected within the SM (red dots in Figure 11b) reflects the greater variability of this mineral with respect to the substitution mechanisms, emphasizing multiple alluvial source components or multiple rutile generations from the same source (i.e., primary vs. secondary—see the discussion in Section 5.1.3, below).

The incorporation of W seems to follow both Equations (2) and (3), whereas the incorporation of $Nb^{5+}$ and $Ta^{5+}$ only follows Equation (5). The excess of $M^{2+}$ and/or $M^{3+}$ cations in some rutile from the samples representing the SM highlights the possibility of other types of substitution, namely protonation ($Ti^{4+} = R^{3+} + H^+$; [64]), or even the existence of anionic vacancies, which typically gives rutile semi-conductor characteristics [98]. The incorporation of trace elements in rutile from metasediments (black dots) is usually more limited. Yet, samples with higher W contents, collected near Sn-W occurrences, mainly follow the trend of Equation (2), revealing their hydrothermal character and formation

under relative reducing conditions (note that in Equation (2), W is paired with $M^{2+}$ cations, most likely $Fe^{2+}$).

The substitution mechanisms in anatase are similar to those in rutile, but much more limited, whereas in brookite the excess of 2+ and 3+ cations predominate, favoring the incorporation of Fe, mainly through mechanisms involving protonation.

### 5.1.3. $TiO_2$ Polymorph's Forming Environments—Genesis and Stability

While the incorporation of trace elements in rutile, anatase, and brookite is largely controlled by intrinsic factors, set by the specific features of each $TiO_2$ polymorph crystal structure, as discussed in Section 5.1.1, it's not less true its dependency on extrinsic parameters, such as the T, P, pH, $fO_2$, and composition (chiefly trace-element availability), of the geological environments in which these minerals form [68,99–104]. Thus, it is crucial to address $TiO_2$ minerals' stability and genesis to better appreciate their compositional variability and assess how different geological contexts can affect them.

$TiO_2$ minerals may have a primary origin, if crystallizing directly from a magma, hydrothermal fluid, or even low-temperature aqueous fluids, or they may have a secondary origin, resulting from the alteration of pre-existing Ti-bearing minerals [61,62,104]. Although $TiO_2$ polymorphs can be formed in all types of rocks (magmatic, metamorphic, or sedimentary), the $TiO_2$ mineral phases present depend on their P-T stability fields [10,68,101–104], rutile being the most stable in most geological conditions. Under magmatic conditions, rutile is the $TiO_2$ mineral to crystallize as it is the only stable phase above 500 °C, while primary anatase and brookite are predominant in authigenic environments, low-T metamorphism, and low-T hydrothermal systems [10,56,62,68,101–104].

At sub-magmatic temperatures, and under low-to-medium-grade metamorphic conditions, Ti is considered immobile [104]. The low solubility of Ti-oxides in metamorphic fluids, implies that host rock $TiO_2$ content is a limiting factor to the amount of $TiO_2$ polymorphs formed during hydrothermal alteration processes [56,64]. Despite its general low solubility under these conditions, Ti can form chloride, fluoride, hydroxide, carbonate, phosphate, and sulfide complexes, increasing its solubility and consequent mobility [105–109]. Hydrothermal rutile and anatase in hydrothermal veins and associated hydrothermal alteration zones in porphyry Cu deposits, e.g., [56,64], are unequivocal proof of Ti mobilization by magmatic-hydrothermal fluids. Although there is increasing evidence supporting the mobilization of Ti via hydrothermal fluids [100,104,110,111], the physicochemical conditions favoring the mobile behavior of Ti have not been fully quantified [56,64].

The weathering of Ti-bearing silicates (e.g., sphene, Ti-rich biotite, Ti-magnetite, and ilmenite) in surficial environments, including alluvia, produces a wide range of secondary minerals, that includes anatase and brookite [112]. After weathering, Ti is initially dissolved, but it quickly precipitates as a hydrated oxide that subsequently crystallizes as anatase, rutile, or brookite [112]. Experimental work on laboratory synthesis of anatase and brookite nano crystals indicates that anatase is the first of the two polymorphs to form due to its low surface energy and that, at temperatures above 600 °C, these two polymorphs transform into rutile [113–115].

Segura metasedimentary rocks are affected by low-grade regional metamorphism, that did not go beyond the chlorite zone of the greenschist facies [77]. Consequently, the minerals formed in this region are mainly governed by T and chemistry. The predominance of anatase and, to a lesser extent, brookite, in the Segura SGC metasediments indicates that these rocks never reached temperatures above 600 °C, otherwise they would have transformed into rutile.

The higher abundance of alluvial rutile within the Segura granites (Figure 6a) most certainly reveals its essential magmatic origin, while the anatase depletion in the SM metamorphic contact halo (Figure 6b) should reflect anatase to rutile conversion, due to a local thermal gradient increase promoted by the emplacement of the SM granites. Following the same reasoning, the positive rutile and negative anatase abundance anomalies associated with the Sn-W mineralized zones (Figure 6a,b) also suggest anatase to rutile

transformation, which is most certainly linked to hydrothermal processes. However, the presence of hidden SM apophysis along this NW–SE trend cannot be ruled out. Since anatase cannot be formed under magmatic conditions, and considering the low mineral dispersion of the area, its occurrence in samples collected within granites most likely expresses either a secondary origin, by the replacement of granite Ti-bearing minerals, and/or a primary origin, via hydrothermal fluid precipitation, both of which are compatible with late magmatic-hydrothermal fluids that characterize most granite-related ore systems.

Although the stability of $TiO_2$ polymorphs seems to be well known, the detailed conditions under which their trace-element substitutions are promoted are still poorly understood [57,62,64,68,99]. Other than high P–T conditions [116–122], the trace element fractionation coefficients between $TiO_2$ polymorphs and fluids relevant to our work (i.e., magmatic and hydrothermal) are unknown. Nevertheless, Mallmann et al. [123] calculated the distribution coefficients for some of the trace elements between rutile and silicate magmas as a function of oxygen fugacity, demonstrating their extreme compatibility with Ta, Nb, W, Cr, and V (Figure 12). Since the distribution coefficients of W, Nb, and Ta between rutile and magma are very similar, the incorporation of these traces depends on magma compositions. The rutile-magma distribution coefficients for Sn are unknown, but its geochemical affinity with Nb and Ta suggests similar behavior.

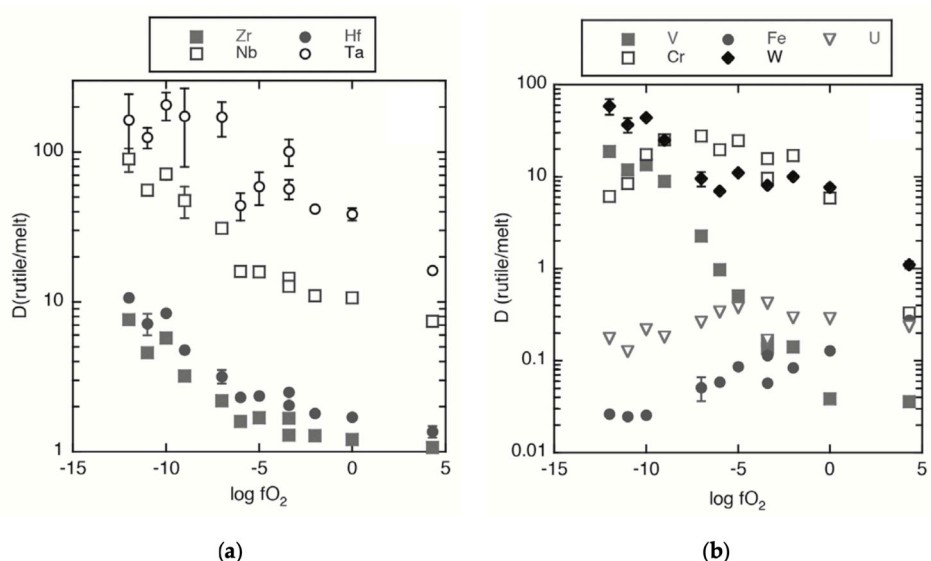

**Figure 12.** Rutile/silicate-melt partition coefficients obtained for HFSEs (**a**) and several heterovalent elements (**b**) plotted against oxygen fugacity [123].

If no common trace element-containing minerals crystallize in magmas, such as specific oxide phases (columbo-tantalite for Nb and Ta, cassiterite for Sn, chromite for Cr, and wolframite or scheelite for W), Fe- and Ti-Fe-oxides, or other minerals, such as sphene, primary magmatic rutile may contain significant concentrations of trace elements if the magmas are enriched in those elements [62]. In fact, some of the highest values in Nb, Ta, and Sn are recorded in rutile from rare metal pegmatites that crystallize directly from hydrated silicate magma [54,57]. Thus, high Sn and W values in magmatic rutile may be used to trace productive granites.

Different overlapping episodes of alteration, acting on contrasting lithologies and different mineral precursors, produce different generations of $TiO_2$ polymorphs [64]. The $TiO_2$ polymorphs generated under such different conditions incorporate different elements in different proportions. It is during these different stages of alteration that rutile incorporates trace elements from both Ti phases and hydrothermal fluids, expanding its chemical composition variability, as observed in this study and in other hydrothermal mineral deposits [49–73]. The trace-element variability within and between crystals, which we also

registered, is therefore a typical characteristic of rutile, resulting in various zoned patterns (Figure 7). The oscillatory zoning records direct crystallization in an open system, mostly reflecting variations in local chemical potentials associated with mineral-growth kinetics (i.e., mineral growth velocity versus elemental chemical diffusion rates) [124], rather than abrupt chemical fluctuations resulting from successive flows of chemically distinct fluids, as suggested by Carrocci et al. [70]. Patchy, overlapping, and more complex zoning patterns disclose the crystallization–recrystallization history of each individual rutile, including interaction with multiple fluids. Zonation patterns were exclusively observed in rutile. The studied specimens of anatase and brookite are quite homogeneous, appearing to be less sensitive to these changes or having a much simpler crystallization history. Note that under authigenic or low-temperature metamorphic conditions, anatase crystals seems to preserve their trace element contents during in situ recrystallization [68,104]. The high anomalous trace element contents of some anatase grains might reflect their direct precipitation from enriched hydrothermal fluids that generated the ore deposits in the region.

*5.2. Trace Element Geochemistry of TiO$_2$ Polymorphs as Tracers of Mineralizing Systems and Its Application to the Exploration for Sn and W Deposits*

The trace-element geochemistry of TiO$_2$ polymorphs, mainly rutile, has been increasingly used to characterize and discriminate different ore-deposit systems [49–73]. However, its application to Sn and W exploration is still limited; it includes rutile from the Panasqueira deposit [70], the largest W deposit in Western Europe, and rutile from the Sn-W Pilok deposit (Thailand) [57].

The alluvial rutile from this study shows chemical zonation patterns (oscillatory, sectoral, patchy, and complex) similar to the rutile crystals from Panasqueira. The maximum W concentration recorded in the alluvial rutile was 7.2 wt.%, slightly lower than the maximum of 10.7 wt.% W reported in Panasqueira, one of the highest in the world [70]. The rutile in Panasqueira is also V-enriched (3.5 wt.% V) compared to that in Segura (1.6 wt.% V). Regarding the remaining trace elements, the rutile in the Segura shows higher maximum concentrations of Nb (12.2 wt.%), Sn (6.7 wt.%), Fe (3.8 wt.%), and Ta (3.7 wt.%) than the Panasqueira rutile (Fe—3.0 wt.%, Nb—1.6 wt.%, Ta—2.2 wt.%; [71]). The high Nb contents in the Segura alluvial rutile reflect its magmatic origin, associated with the highly differentiated and rare metal specialization of the SM granites [82].

The average W concentration of alluvial rutile from Segura (0.66 wt.% W) is comparable to that of rutile from the Pilok deposit (0.72 wt.% W). However, the mean values of Nb, Ta, and Sn (7.15 wt.%, 1.88 wt.%, and 1.72 wt.%, respectively) from this Asian deposit [57] are higher than the mean values of the same elements in the studied rutile (0.82 wt.% Nb, 0.13 wt.% Ta, and 0.57 wt.% Sn). It should be noted that the values published for Pilok represent rutile from sub-mineralized zones of the deposit, unlike the present work, in which it was not possible to have this level of control over the rutile sources.

Although the Minnie Springs Prospect in Western Australia is a Mo-mineralization and not a Sn- and/or W-deposit, it is still a granite-related ore system, and [68] is one of the few papers to have analyzed the three TiO$_2$ polymorphs to assess their potential as exploration tools. Plavsa et al. [68] distinguished rutile according to its mineralized/non-mineralized sample character. Rutile from non-mineralized samples, have lower trace element contents than rutile from Segura, except for Cr and Fe. Rutile from mineralized samples have similar maximum Nb and Ta concentrations (109,142 ppm Nb and 17,815 ppm Ta) to rutile from Segura (85,347 ppm Nb and 19,139 ppm Ta). The maximum Fe content in rutile from the Australian mineralized samples are much higher (50,490 ppm) than that of rutile from Segura (29,258 ppm). Maximum V, Sn, and W concentrations in Segura rutile (V—10,937 ppm; Sn—55,665 ppm; W—57,173 ppm) are higher than the Minnie Springs Prospect rutile (V—4210 ppm; Sn—2590 ppm; W—23,620 ppm), as would be expected considering the contrast between the Segura Sn-W system and the Mo system of the Australian mineralization. For this comparison, the much higher values in Sn and W of anatase from Segura are worth highlighting (Segura: Sn—4892 ppm; W—10,523 ppm;

Minnie Springs: Sn—170 ppm; W—830 ppm). The maximum values of Nb in anatase are similar, while in Segura V and Ta are higher and Fe is lower. Brookite in the Australian mineralization was only identified in the non-mineralized samples and show higher Nb and Fe and lower V, Cr, Sn, Ta, and W contents than the brookite in our study. Again, Sn (6330 ppm) and W (3457 ppm) are much higher in brookite from Segura than those from Minnie springs (Sn—53 ppm; W—190 ppm). The Fe content of the Australian brookite is considerable, like in Segura, reflecting their secondary origin by alteration of Fe and Ti minerals.

A triangular Ti-100(Fe+Cr+V)-1000(Sn+W) discriminant diagram has been used to distinguish rutile signatures from mineralized and non-mineralized samples [57]. Most rutile from rocks unaffected by mineralizing hydrothermal fluid plot along, or near, the Ti-(Fe+Cr+V) axis [58], while rutile associated with mineralizing metamorphic/metasedimentary processes tend to approach the Sn+W vertex. Rutile and anatase in our study were plotted on this diagram and on the (Nb+Ta)-Sn-W diagram (Figure 13). Due to brookite scarcity and their low trace element content, brookite was not plotted. We plotted samples from HM assemblages representing granites (Figure 13a) and the different mineralized vein types (Figure 13b). Within granite assemblages we have distinguished three groups: border zone, internal zone, and aplite-rich zone.

In the Ti-100(Fe+Cr+V)-1000(Sn+W) diagram rutile, the dominant $TiO_2$ mineral in granites, plots along the mineralizing trend defined by Clark and Williams-Jones [57]. In the (Nb+Ta)-Sn-W diagrams of HM granite assemblages it is possible to discriminate a Sn-enriched trend, in samples from the innermost zone of the SM, and a W-enriched trend in the SM border zone. These two trends may reflect primary magmatic-hydrothermal compositions representing the two main magmatic pulses recognized by the two mica granite core facies, and the muscovite granite border facies. The aplite-rich zone shows rutile compositions that fall into both trends. Note that samples from this group were collected in an inward-outward trend, thus including both granitic facies, plus the aplitic contributions. Rutile compositions plotted outside of these trends are probably from secondary rutile grains that formed from the breakdown of other magmatic Ti-bearing mineral phases.

An enrichment in Sn+W is unequivocally perceptible in rutile from the Li-Sn and W-Sn HM associations, with some dispersed compositions (including a few trace element-free grains), again corresponding to rutile populations with distinct sources and origins. The most enriched rutile population should represent grains that formed under mineralizing processes, whereas the others represent more regional metamorphic processes. Also, note the depletion of Nb+Ta relative to Sn and W of these hydrothermal rutile when compared with magmatic ones, suggesting precipitation from a fluid with lower (Nb+Ta)/(W+Sn) ratios than the magma. Also, this ratio is somewhat constant in rutile from the W-Sn HM association, with data projecting parallel to the Sn-W axis (80%–90%). These samples show appreciable amounts of wolframite, suggesting that they represent W-dominated quartz veins. Rutile from the lithiniferous aplite-pegmatite HM assemblages show two populations, one with higher Sn and another with higher Nb+Ta. In the Sn-W systems, in which cassiterite is the dominant oxide, the Sn contents of the few analyzed rutile grains are lower, supporting the hypothesis that during cassiterite co-precipitation, the incorporation of Sn into the rutile is inhibited.

Anatase shows a characteristic trend toward the Sn+W vertex in the triangular Ti-(Fe+Cr+V)-(Sn+W) diagrams, illustrating an enrichment, although not as pronounced as rutile in the corresponding samples. This trend essentially reflects a relative enrichment in W, as can be seen in the Sn-(Nb+Ta)-W diagrams, and is probably linked to mineralizing hydrothermal processes, thus showing that anatase may also correlate with mineralization/alteration zones.

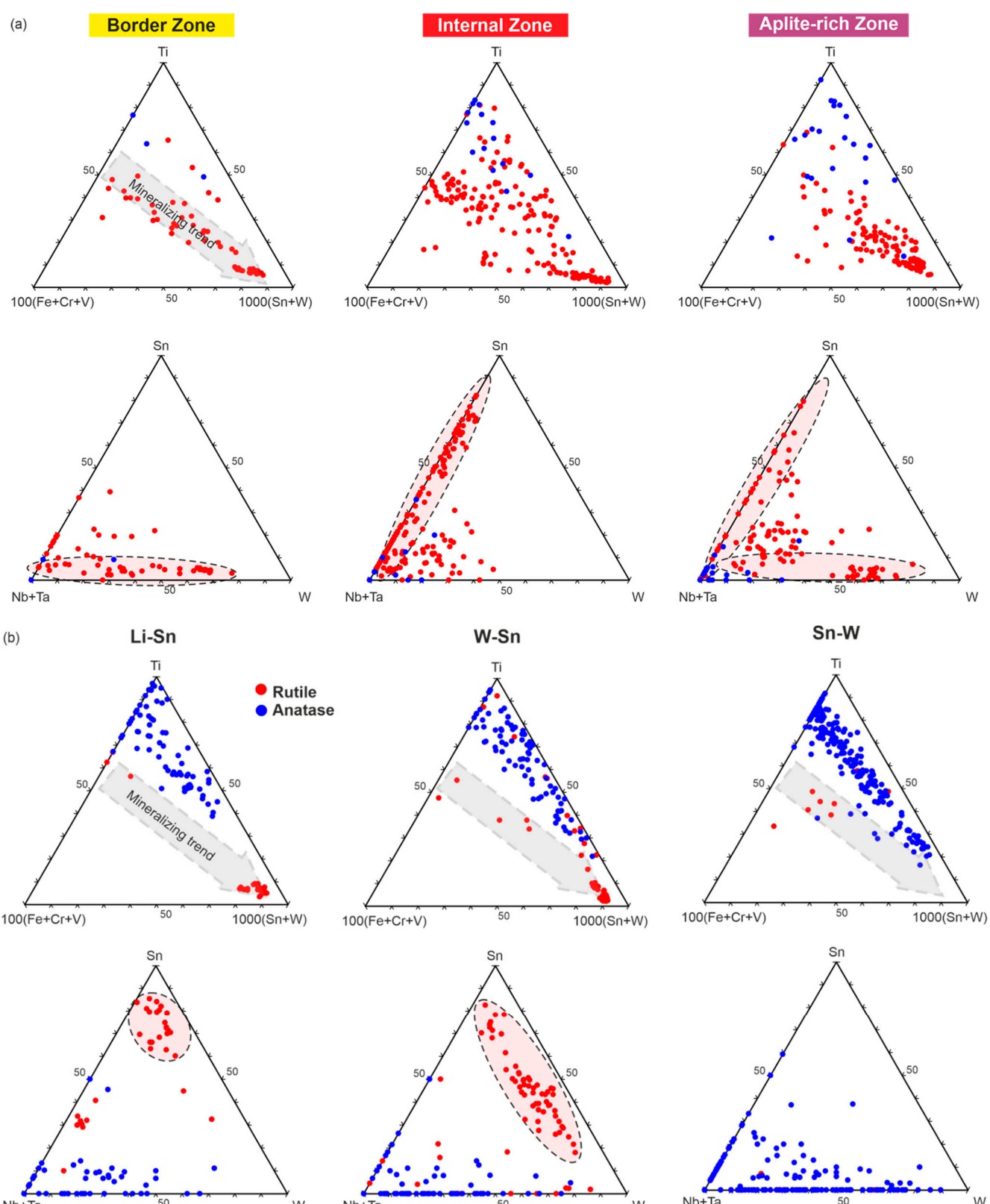

**Figure 13.** Triangular Ti-100(Fe+Cr+V)-1000(Sn+W) and Sn-(Nb+Ta)-W discriminant diagrams for rutile and anatase trace-element composition [57,68]. Compositions of rutile and anatase from: Granite HM assemblages, considering 3 groups—border zone, internal zone, and aplite-rich zone (**a**); HM assemblages for distinct granite-related mineralizing systems (**b**). Adapted from [37].

In the search for exploration footprints, IDW interpolated maps for Sn, W, Nb, Ta, Fe, and V compositions of rutile and anatase were prepared, following the same methodology as in the HM abundance maps (Section 4.1.1), using the ppm median values of each sample instead. The scarcity of brookite and its essential secondary origin were enough reasons to exclude it from this analysis.

Sn and W show similar behavior in rutile (Figure 14). Both elements show marked positive anomalies in the vicinity of the Li-Sn aplite–pegmatites, close to the W-Sn veins and accompanying the SM. Antunes et al. [83] defined SM granites as being enriched in Sn and directly related to Sn-W veins. In this context, the presence of abnormally Sn-enriched rutile, taken as magmatic, is a good indicator of specialized magmas for Sn and an excellent guide for exploration. When carefully observed, a slight difference in the behavior of these two elements can be spotted in the SM outcropping area, with a more intense positive Sn anomaly in the innermost zones, but not in W. This observation is also in line with the presence of more than one magmatic pulse, reflected in different granitic facies, and in the distinct populations shown by the discriminant triangular diagrams discussed above, corroborating the hypothesis of some W specialized facies. Particular attention should be paid to very pronounced and defined negative anomalies in the areas dominated by Sn-W veins. These probably indicate the co-precipitation of rutile and cassiterite, with the latter trapping the available Sn, leaving rutile depleted in this element. In samples close to the cassiterite-bearing lithiniferous aplite-pegmatites the anomaly is positive. In this case, either the minute cassiterite in these samples did not remove all the Sn from the system, being able to incorporate rutile, or rutile precedes cassiterite crystallization. In a given region, the presence of Sn-depleted rutile does not mean the absence of mineralization, if Sn-enriched rutile are found in the area; and a negative Sn anomaly can even be an excellent footprint of cassiterite-rich deposits, especially if combined with other trace element analyses. In areas with W-Sn veins, the presence of Sn- and W-enriched rutile suggests rutile precipitation preceding wolframite and/or scheelite deposition, requiring confirmation in future work.

The behavior of Sn and W in anatase is somewhat different from rutile (Figure 14). Anatase shows positive Sn and W anomalies in the Segura Massif edge zone and in the vicinity of the wolframite-rich veins, but also close to the Sn-W veins. It must be taken into consideration that despite this latter positive anomaly, Sn content in anatase is only slightly higher than Sn content in rutile from the same area. Considering that anatase does not incorporate trace elements as efficiently as rutile, the pronounced positive W anomaly observed in association with W-Sn mineralized zones indicates that the metamorphic processes responsible for wolframite mineralization made enough W available in the system to enable its incorporation in the $TiO_2$ polymorphs. The Sn and W behavior in anatase shows that rutile is not the only $TiO_2$ polymorph in which signs of Sn and W mineralization are recognized.

The rutile's positive Nb and Ta anomalies are clearly associated with the SM granitic rocks (Figure 14). The Nb and Ta enrichment in rutile is typically a result of magmatic to late magmatic-hydrothermal processes since they are rarely mobile elements in most hydrothermal or metamorphic conditions [57]. The enrichment of these elements in rutile associated with granitic rocks is less surprising if we consider the strong compatibility of Nb and Ta with rutile, exhibiting distribution coefficient values between 10 and greater than 100 (Figure 12) [123].

The behavior of Nb, Ta and Fe in anatase is similar to that in rutile, with the same notable positive anomaly in the SM (Figure 14), most likely reflecting its secondary character.

The behavior of Fe in the rutile is like that of Nb and Ta, with a positive anomaly in SM outcrops, suggesting a common origin. However, on the remaining map Fe values are high compared to Nb and Ta, with small positive anomalies around the Li-Sn aplite–pegmatites and close to the W-Sn-dominated quartz veins. This difference can be explained by the higher mobility of Fe compared to Nb and Ta in hydrothermal and metamorphic environments, as well as by the inherited composition of its precursors in the case of a secondary origin.

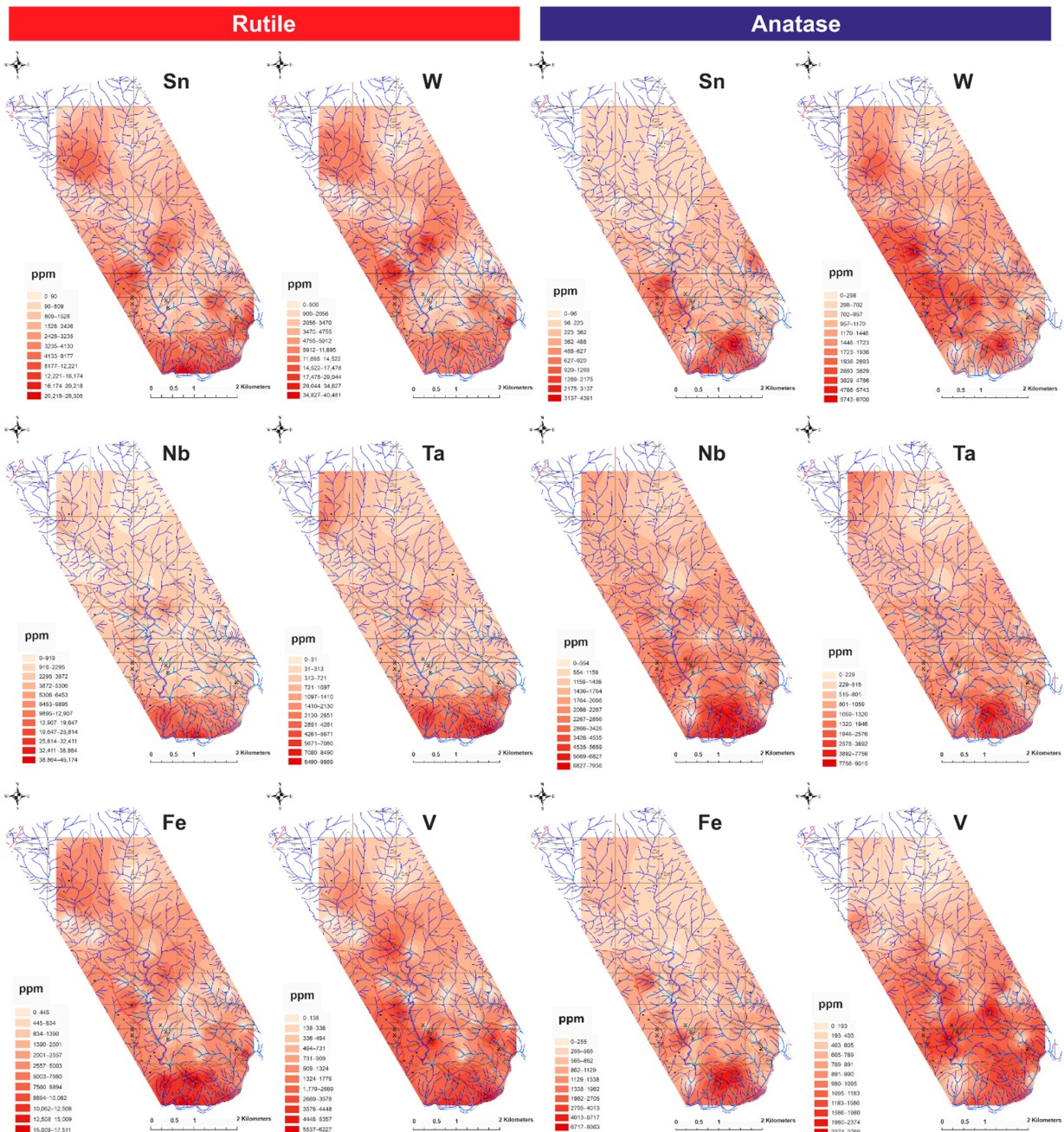

**Figure 14.** Trace element abundance distribution maps for alluvial rutile and anatase. Inverse distance weighted interpolated maps created with ArcGIS software [37].

The V in rutile shows a very interesting behavior, since the small positive anomalies identified are associated with all the mineralized zones (Sn-W, W-Sn, and Li-Sn) and the SM granites (Figure 14). To clearly demonstrate that the V content was enhanced by the alteration processes associated with the mineralization [57] it would be necessary to determine the regional background V values in rutile. These results could lead to the conclusion that V is also a good marker of hydrothermal and/or magmatic processes.

As with rutile, anatase also reveals a small positive V anomaly near the Sn-W mineralized structures (Figure 14). The anomalies in the zone of the SM and close to the Li-Sn aplite-pegmatites are not as pronounced, nor are those N of the Li-Sn aplite-pegmatites, near a granodiorite porphyry dyke.

The IDW maps of the alluvial $TiO_2$ minerals' trace element composition needs further refinement to fill the gaps caused by limited data and to extend the area around the SM. However, the preliminary results are consistent enough to be used as a footprint for Sn and W deposits.

## 6. Concluding Remarks

This work documented the ubiquitous presence of rutile and anatase, and the low abundance of brookite, in the non-magnetic fraction of alluvial-heavy-mineral concentrates from Segura. Alluvial HM analysis proved to be a good tool for identifying the contribution of detrital minerals to the alluvium, including major outcropping lithotypes and mineralized bodies, supporting the immense potential of HM mineral assemblages as a proxy for the local geology. The data on alluvial $TiO_2$ polymorphs' relative abundance and the respective IDW maps reveal the immense potential of recognizing the metamorphic/metasedimentary processes related to the emplacement of Sn-W productive intrusive bodies. The IDW abundance maps for alluvial cassiterite, wolframite and scheelite should themselves signal the presence of mineralized bodies.

The trace element geochemistry of Segura alluvial rutile, anatase, and brookite is highly variable, implying multiple sources and a diversity of mineral-forming processes. The main compositional differences observed among the $TiO_2$ polymorphs resulted from the disparities in their crystal structures and related intrinsic factors, as well as from variations in the extrinsic parameters characteristic of their forming environment (P, T, and X). Substantial anomalous Sn, W, Nb, Ta, Fe, V, and Cr enrichments were registered, particularly in rutile but also in anatase. Brookite usually has low trace element contents, except for Fe. Based on these variations, it was possible to identify several populations of rutile and anatase linked to Sn-W mineralizing processes, showing that rutile is not the only $TiO_2$ polymorph in which these signs are recognized. Some HFSE-rich and granitophile-rich rutile are most likely magmatic, forming in extremely differentiated melts. Their Sn and W contents enable discrimination between Sn-dominant and W-dominant systems. In addition to magmatic rutile, other trace element-enriched rutile and anatase are believed to be associated with mineralizing hydrothermal fluids related to Sn-W, W-Sn and Li-Sn systems.

Trace element distribution IDW maps for rutile and anatase enabled the identification of both positive and negative geochemical anomalies, which can be linked to the main local mineral occurrences. An appropriate interpretation of these anomalies requires a good understanding of the $TiO_2$ mineral-containing paragenetic sequence, i.e., the precipitation timing of $TiO_2$ polymorphs relative to Sn and W ore minerals.

Despite the complexity of each alluvial sample and emphasizing the need for a complete and unambiguous distinction between $TiO_2$ polymorphs, this contribution recognizes that heavy mineral analysis combined with trace element geochemistry of alluvial $TiO_2$ polymorphs can be a robust, cost- and time-effective, exploration tool for Sn(W) and W(Sn) ore deposit systems.

**Supplementary Materials:** The following supporting information can be downloaded at: https://www.mdpi.com/article/10.3390/min12101248/s1. Figure S1: Animated 3D rutile structure [96]; Figure S2: Animated 3D anatase structure [96]; Figure S3: Animated 3D brookite structure [96]; Table S1: Representative rutile EPMA compositional data; Table S2-Representative anatase EPMA compositional data; Table S3 Representative brookite EPMA compositional data.

**Author Contributions:** This work is part of the second author (N.G.)'s MSc thesis, under the supervision of the first (M.G.) and third (R.S.) authors. Conceptualization, M.G. and R.S.; heavy-mineral analysis, R.S. and N.G.; sample preparation, N.G.; EPMA, M.G. and N.G.; micro-Raman, M.C.; numerical handling of analytical data, N.G. and M.G.; data interpretation, M.G., N.G. and R.S.; writing—original draft preparation, M.G.; writing—review and editing, R.S., N.G. and M.C.; funding M.G., R.S. and M.C. All authors have read and agreed to the published version of the manuscript.

**Funding:** This work was supported by Fundação para a Ciência e Tecnologia, I.P./MCTES through national funds, (PID-DAC)–UIDB/50019/2020 (IDL), UIDB/04449/2020, and UIDP/04449/2020 (HERCULES Laboratory), as well as by the MOSTMEG project (ERA-MIN/0002/2019), http://mostmeg.rd.ciencias.ulisboa.pt/ (accessed on 21 September 2022).

**Acknowledgments:** The authors thank the previous research projects: (i) Inventariação e prospecção de terras raras nas regiões fronteiriças da Beira Baixa e do Norte Alentejo, funded by the European Union Cross-border INTERREG II program; and (ii) Prospecção de Estanho e Volfrâmio e outros metais associados-Faixa Gois-Segura, funded by national funds, PID-DAC, for the sampling, treatment, and partial study of the alluvial samples. Furthermore, thanks are due to Catarina Moniz and Pedro Patinha, both from LNEG, for helping in the vectorial geological mapping of the study area. The constructive comments and suggestions of the reviewers were much appreciated.

**Conflicts of Interest:** The authors declare no conflict of interest.

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
