# Peer review of "Trace Element Geochemistry of Alluvial TiO2 Polymorphs as a Proxy for Sn and W Deposits"

_minerals, doi:10.3390/min12101248_

Round 1
Reviewer 1 Report
The manuscript submitted by Gaspar et al. reports the trace element geochemistry of alluvial TiO2 polymorphs, which can be used as a proxy for Sn & W deposits. Overall, this paper is well-written, containing much useful information, thus it can be accepted for publication. I think only minor revision is needed. Please see my comments below.
At the beginning of the abstract, the significance of this research should be addressed, i.e., what is the scientific issue of the study area? Why the researchers should carry out such a study?
Line 38-40: one reference is not enough. Add more., For example: Li, H., Zhu, D.P., Shen, L.W., Algeo, T.J., Elatikpo, S.M., 2022. A general ore formation model for metasediment-hosted Sb-(Au-W) mineralization of the Woxi and Banxi deposits in South China. Chemical Geology, https://doi.org/10.1016/j.chemgeo.2022.121020; Li, H., Zhu, D.P., Algeo, T.J., Li, M., Jiang, W.C., Chen, S.F., Elatikpo, S.M., 2022. Pyrite trace element and S-Pb isotopic evidence for contrasting sources of metals and ligands during superimposed hydrothermal events in the Dongping gold deposit, North China. Mineralium Deposita, DOI:10.1007/s00126-022-01128-w; Zhu, D.P., Li, H., Algeo, T.J., Jiang, W.C., Wang, C., 2021. The prograde-to-retrograde evolution of the Huangshaping skarn deposit (Nanling Range, South China). Mineralium Deposita 56, 1087–1110.
Line 41-44: also need references.
Line 37-69: the authors need to address the scientific issue of the study area, i.e., Why the researchers should carry out such a study?
Huan Li
Central South University, China
Author Response
The point raised by the reviewer to clarify the readers by addressing the significance of this research in both Abstract and Introduction is relevant and was taken into account.
Relative to the reference issues we understand the reviewer criticisms but we would like to point:
1 - Citation of a reference book seems sufficient to address general concepts. Of course we can find multiple examples, and the ones relevant to this work are further presented in more detail, either as individual and group citations along the introduction chapter or along the text when needed.
2- For this part of the text (Line 41-44) "Trace element distribution in mineralized rocks and corresponding geochemical alteration halos, is nowadays critical for understanding the mineralizing processes and to evaluate the economic potential of a particular occurrence or region." is more a rectorial sentence expressing the authors opinion based on their accumulated experience that is also accepted as common knowledge, within and outside of the scientific community. Examples and citations, as addressed in the previous point, are presented throughout the manuscript.
Reviewer 2 Report
1. Summary
Segura alluvial samples and geochemistry of their TiO2 polymorphs (rutile, anatase, and brookite) were investigated for their potential as an exploration tool for Sn- and W-deposits.
Trace element geochemistry of Segura alluvial rutile, anatase, and brookite was highly variable, implying multiple sources and a diversity of mineral-forming processes. Main compositional differences among TiO2 polymorphs are related with intrinsic (structural) factors, and to P-T-X extrinsic parameters of their forming environments.
2. General comments
The article is clear, well organized, and correctly written and presents a well-designed methodology in accordance with the objective to be achieved.
3. Some questions
Has the methodology used, i.e., the geochemical analysis of titanium polymorphs, already been used in other W-Sn mineralized areas, in Portugal or outside Portugal?
Alluvial rutile from this study is compared with rutile crystals from Panasqueira. Are there similarities with other W-Sn deposits besides Panasqueira?
The authors point out that trace element distribution IDW maps for rutile and anatase enabled the identification of both positive and negative geochemical anomalies that can be linked to the main local mineral occurrences.
Couldn't some geostatistical analysis be used in the treatment of this data besides the comparison of trace element distribution IDW maps?
Author Response
The main comments/suggestions and questions done by the Reviewer are addressed point by point as follows:
1- Our understanding is that the Reviewer suggestion was to cut the aspects related with the heavy mineral study main conclusions from the abstract/summary and mainly focus on the TiO2 trace element geochemistry. We understand that the abstract must be short and concise and that it was a bit long. However, this issue is relevant to the geochemistry interpretations. Since we included an initial sentence to better clarify the objectives and significance of the study area, as suggested by the other Reviewer, we decided to rephrase and shorten the HM study in the summary to the minimum but necessary points.
2- We acknowledge the general positive appraisal of the manuscript.
3- The response to the 3 questions raised by the Reviewer are as follows:
Q -"Has the methodology used, i.e., the geochemical analysis of titanium polymorphs, already been used in other W-Sn mineralized areas, in Portugal or outside Portugal?"
A - To our knowledge the only geochemical analysis of titanium polymorphs associated with W-Sn deposits are the ones addressed in lines 74-81 of the Introduction section, as well as minor and incomplete contributions in unpublished MSc and PhD thesis.
Q - "Alluvial rutile from this study is compared with rutile crystals from Panasqueira. Are there similarities with other W-Sn deposits besides Panasqueira?"
A - By answering the previous question is obvious that published data is scarce and comparison was done with the published results of our knowledge, not only with Panasqueira but also with rutile from the Sn-W Pilok deposit, as well as with the Mo Minnie Springs Prospect in Western Australia at the beginning of section 5.1.
Q - "The authors point out that trace element distribution IDW maps for rutile and anatase enabled the identification of both positive and negative geochemical anomalies that can be linked to the main local mineral occurrences. Couldn't some geostatistical analysis be used in the treatment of this data besides the comparison of trace element distribution IDW maps?"
A - We understand the power of geostatistical analysis and that approach is planned after filling some data gaps and extending the study area in research projects that follow this manuscript, enabling the development of a much more robust exploration model.